

# An IBBCEAS system for atmospheric measurements of glyoxal and methylglyoxal in the presence of high NO₂ concentrations

Jingwei Liu[1], Xin Li[1,2], Yiming Yang[1], Haichao Wang[1], Yusheng Wu[1,*], Mindong Chen[2], Jianlin Hu[2], Xiaobo Fan[3], Limin Zeng[1], and Yuanhang Zhang[1]

[1] State Key Joint Laboratory of Environmental Simulation and Pollution Control, College of Environmental Sciences and Engineering, Peking University, Beijing, 100871, China P. R.
[2] Collaborative Innovation Centre of Atmospheric Environment and Equipment Technology, Nanjing University of Information Science & Technology, Nanjing, 210044, China P. R.
[3] Quadrant Space (Tianjin) Technology Co., LTD, Tianjin, 301700, China P.R.
[*] Now at: Department of Physics, University of Helsinki, Helsinki, 00014, Finland

*Correspondence to*: Xin Li (li_xin@pku.edu.cn)

**Abstract.** A system based on incoherent broadband cavity enhanced absorption spectroscopy (IBBCEAS) has been developed for simultaneous measurement of nitrogen dioxide (NO₂), glyoxal (GLY) and methylglyoxal (MGLY). On this system, the absorption of light around 460 nm is spectrally resolved. The concentration of absorbers is determined from a multi-component fit. At an integration time of 100 s, the measurement sensitivity (2σ) for NO₂, GLY, and MGLY can reach 18 ppt, 30 ppt, and 100 ppt, respectively. The measurement uncertainty which mainly originates from path length calibration, sampling loss, and uncertainty of absorption cross sections is estimated to be 8% for NO₂, 8% for GLY, and 16% for MGLY. When applying the instrument during field observations, we found significant influence of NO₂ on spectra fitting for retrieving GLY and MGLY concentration, which is caused by the fact that NO₂ has higher absorption cross section and higher ambient concentration. In order to minimize such an effect, a NO₂ photolytic convertor (NPC) which removes sampled NO₂ at an efficiency of 76% was integrated on the IBBCEAS system. Since sampled GLY and MGLY are mostly (⩾ 95%) conserved after passing through the NPC, the quality of the spectra fitting and the measurement accuracy of ambient GLY and MGLY were largely improved.

## 1 Introduction

Glyoxal (CHOCHO, GLY) and methylglyoxal (CH₃COCHO, MGLY) are typical atmospheric α-dicarbonyl species that primarily enter the atmosphere through direct emissions from biomass burning and the oxidation of volatile organic compounds such as isoprene, aromatics, and alkenes (Fu et al., 2008). Therefore, GLY and MGLY are suitable indicators of the atmospheric oxidation capacity (DiGangi et al., 2012). Atmospheric sinks of GLY and MGLY include photolysis and reactions with OH radicals (Volkamer et al., 2005a; Fu et al., 2008), which play an important role in the formation of both O₃ and peroxyacetyl nitrate (PAN) (Müller et al., 2016). Furthermore, the contribution of GLY and MGLY to the formation of





second organic aerosol (SOA) has drawn widespread attention in the past few years (Washenfelder et al., 2011; Nakao et al., 2012; Meng et al., 2018). Although GLY and MGLY have relatively low molecular weights, they can form oligomers and participate in SOA formation in aqueous particles(Yu et al., 2011; Hamilton et al., 2013). In order to develop an in-depth understanding of the above processes, fast online measurements of GLY and MGLY with good sensitivity and accuracy are

required.

Techniques for online measurements of GLY and MGLY can typically be categorized as methods based on either mass spectrometry or absorption spectroscopy. While proton transfer reaction time-of-flight mass spectrometry (PTR-ToF-MS) is considered to be a good mass spectrometry technique for measuring VOCs, its sensitivity is still too low to monitor the ambient concentration of GLY (Thalman et al., 2015; Stonner et al., 2017) and both $(H_2O)_3H_3O^+$ and acrylic acid $(C_3H_4O_2)$

can interfere with MGLY results because they have the same $m/z$ ratio (Thalman et al., 2015; Yuan et al., 2017; Zarzana et al., 2018). There are also many types of optical methods, including long path differential optical absorption spectroscopy (LP-DOAS) (Volkamer et al., 2005a), cavity-enhanced differential optical absorption spectroscopy (CE-DOAS) (Thalman and Volkamer, 2010), multi-axis differential optical absorption spectroscopy (MAX-DOAS) (Li et al., 2014), laser-induced phosphorescence (LIP) (Henry et al., 2012), incoherent broadband cavity enhanced absorption spectroscopy (IBBCEAS)

(Min et al., 2016), and methods using satellite techniques like the global ozone monitoring experiment (GOME) (Vrekoussis et al., 2010) and ozone monitoring instrument (OMI) (Miller et al., 2014).

Since the first application of IBBCEAS by Fiedler (Fiedler et al., 2003), it has been widely used in laboratory studies and field campaigns to measure atmospheric trace gases such as $H_2O$, $O_3$, $O_4$, IO, $I_2$, OIO, $SO_2$, $NO_2$, $NO_3$, $N_2O_5$, HONO, HCHO, GLY, and MGLY (Vaughan et al., 2008; Washenfelder et al., 2008; Thalman and Volkamer, 2010; Axson et al., 2011;

Kahan et al., 2012; Min et al., 2016; Wang et al., 2017; Duan et al., 2018). The core part of the instrument is a high-finesse cavity that holds of a pair of high-reflectivity mirrors, which typically have reflectivity greater than 0.9999. Because of the unmodulated broadband light source and multichannel detector, the concentrations of trace gases that have absorption structures can be determined simultaneously. Washenfelder et al. were the first to use this technology to measure GLY and MGLY, and their IBBCEAS instrument has been used for ground-based field campaigns and aircraft measurements

(Washenfelder et al., 2008; Min et al., 2016). Within the range where GLY and MGLY have absorption structures, $NO_2$, $H_2O$, and $O_4$ also have structured absorptions; the spectra fitting and resultant concentrations of these α-dicarbonyl species may have interferences from $NO_2$ because $NO_2$ has a strong absorption structure between 430 nm to 470 nm and the ambient concentration of $NO_2$ is much higher than those of GLY and MGLY. Thalman et al. first encountered the challenge of fitting GLY and MGLY absorption spectra in the presence of high $NO_2$ concentrations (Thalman et al., 2015). To our knowledge,

an effective method has not yet to solve this problem.

In this study, we present an incoherent broadband cavity enhanced absorption spectroscopy system for measuring GLY and MGLY and we describe the instrumental setup in detail. Instrumental sample loss was systematically determined using a novel, self-developed standard gas generator to supply GLY and MGLY and by combining our IBBCEAS with a $NO_2$ photolytic converter developed in-house, laboratory tests and in situ measurements were performed to investigate the



interference of NO$_2$ on spectra fitting and measurements of GLY and MGLY. The accuracy of GLY and MGLY measurements in the presence of high NO$_2$ concentrations are discussed in terms of both experimental results and spectral simulations.

## 2 Instrumental setup

Instruments used in this study include a multi-gas calibrator (146i, Thermo Fisher Scientific Inc., Waltham, MA, USA), a standard gas generator for GLY and MGLY, a NO$_2$ photolytic converter (NPC), and a IBBCEAS. The first two instruments were used to supply and maintain a constant concentration of gas mixture, i.e., NO$_2$ and either GLY or MGLY. The NPC was used to photolyze the majority of the NO$_2$ in the mixed gas and the IBBCEAS was used as the detector for the three gases. Besides the multi-gas calibrator, the other three instruments were developed in our laboratory. The optical layout,
flow system, and operation of our IBBCEAS are described in Sect. 2.1 and the details of standard gas generator and NPC are described in Sects. 2.2 and 2.3.

### 2.1 IBBCEAS

#### 2.1.1 Optical layout

The optical layout of the IBBCEAS system consists of a light source module, an optical cavity module, and a detection
module. The core of the light source module is a single-color LED (M450D3, Thorlabs, Newton, NJ, USA), centered at 445 nm with a full width at half maximum (FWHM) of 18 nm. The LED is powered by a constant current source and fixed on a heat sink connected to a thermostat in order to minimize fluctuations of its operating current and operating temperature. By setting appropriate PID (Proportion Integration Differentiation) parameters of the thermostat, the operating temperature were stabilized at 27.0 ± 0.1 °C, in order to reduce intensity drift and wavelength shift of the LED.
A schematic depiction of the main body of the optical cavity module, which consists of seven mounting plates and four stainless steel bearings, is shown in Fig. 1a. The mounting plates are used to fix optical components and the bearings are used to ensure the coaxiality of these components. The entire cavity module adopts the cage structure design to enhance the system stability; this design is more convenient for replacing parts and adjusting the optical path, which can improve system reproducibility. Light emitted from the LED is directed into the cavity module by an optical fiber (1606205, Avantes,
Apeldoorn, The Netherlands) and before coupled to the cavity, the light is collimated by an achromatic lens (ACA254-030-A, Thorlabs, Newton, NJ, USA) with a 30.0 mm focal length, which is mounted on a XY translating mount (CXY1, Thorlabs, Newton, NJ, USA) to finely adjust its position in both the horizontal and vertical directions. Parallel light behind the lens is introduced into a high-finesse cavity formed by a pair of high-reflectivity mirrors (122330, Layertec GmbH, Mellingen, Germany) with a radius curvature of 1.0 m and a diameter of 25.0 mm. The distance between the mirrors is 84 cm and the
reflectivity of the mirrors, which are each fixed on a customized adjusting rack to finely adjust their pitch and yaw, is reported to be greater than 0.9998 between 420 nm and 450 nm. There is a small hole in the adjusting rack for the purge gas



to pass through and we typically use high purity nitrogen (> 99.999%) as the purge gas to protect the surface of mirrors. After passing through the cavity, light is focused by another achromatic lens (ACA254-050-A, Thorlabs, Newton, NJ, USA) with a 50.0 mm focal length and stray light behind this lens is eliminated by a bandpass filter (FF01-442/42-25, Semrock, New York, USA).

Light exits the cavity module through a fiber bundle (SR-OPT-8024, Andor Tech., Oxford Instruments, England) that is coupled to the detection module, which is a grating spectrometer with a charge-coupled device (CCD) detector (Shamrock 303i, Andor Tech., Oxford Instruments, England). The system uses 600 l·mm$^{-1}$ diffraction gratings (500 nm blaze) centered at 450 nm with wavelength coverage from 380 nm to 519 nm; the width of entrance slit is 100 μm and the corresponding wavelength resolution is 0.47 nm, which is determined by fitting the narrow emission line of a mercury lamp (Hg-1, Ocean

Optics, Dunedin, FL, USA) at 435.84 nm.  When operating the CCD detector, it is cooled to -70 ℃ to prevent noise generated by dark current.

### 2.1.2 Flow system

As shown in Fig. 1, the IBBCEAS flow system includes an inlet tube, aerosol filter, optical cavity, temperature sensor, pressure sensor, rotary vane pump (50104, Thomas, Gardner Denver, Germany), rotameter, three mass flow controllers

(MFCs), and three solenoid valves (6014, Bürkert, Ingelfingen, Germany). Since Teflon has the best GLY transmission efficiency (Min et al., 2016), the inlet tube, aerosol filter and optical cavity were all constructed from Teflon.

After ambient air enters the IBBCEAS system through the fluorinated ethylene propylene (FEP) inlet tube, there is a polytetrafluoroethylene (PTFE) filter (25 μm thickness, 4.6 cm diameter, 2.5 μm pore size, Typris, China) to remove ambient aerosols. Gas passed through the filter is directed into the PTFE optical cavity (40.0 mm O.D., 20.0 mm I.D.) and its

temperature and pressure are measured by the sensors after the cavity. The mass flow controller and a rotary vane pump at the end of the flow system maintain a constant gas flow rate through the cavity: 2 L/min.

The solenoid valve separates a by-pass line, which includes a rotameter, that branches from the inlet tube in front of the PTFE filter; this by-pass line is closed unless a reference spectrum is to be measured. The two remaining MFCs are used to control the flow rate of the two purge lines, through which either nitrogen or helium can enter the cavity module depending

on which line has been opened by the solenoid valve. All valves and MFC flow rates are set automatically. Further operational details are given in the following sections.

### 2.1.3 Operation

The operation of the IBBCEAS system can be divided into four working modes, as shown in Fig. 1: measuring the spectrum of nitrogen (N$_2$ mode), measuring the spectrum of helium (He mode), measuring reference spectra (Reference mode), and

measuring sample spectra (Sample mode). The first two working modes are used to calculate the mirror reflectivity and the other two are used to calculate the concentrations of trace gases. The theoretical equations used for these calculations are given in Sect. 3.



The intensity of measured spectra take some time to stabilize when the gas in the cavity is switched from nitrogen to helium. The amount of time for intensity stabilization is inversely related to the helium flow rate into the cavity; we investigated helium flow rates between 100 mL/min to 500 mL/min and measured the intensity at 450 nm (Fig. 2a). In order to reduce signal stabilization time and minimize signal fluctuation, we set the flow rate to 400 mL/min as it takes 2 minutes to achieve

a stable signal. The signal also needs time to stabilize when the gas in the cavity is switched from the reference gas to sample air; we investigated a series of $NO_2$ concentration gradients as sample air to quantify this time. The results shown in Fig. 2b indicate that at least 20 s are required for different $NO_2$ concentrations to reach steady state; therefore, we purged the cavity for 20 s when the system was switched between Reference mode and Sample mode. When operating our IBBCEAS system with the above settings, it typically takes 5 minutes to calibrate the reflectivity of the mirrors each day (2 min in $N_2$ mode, 3

min in He mode), 2 min per hour to measure reference spectra in Reference mode, and for the rest of the day the system is operated in Sample mode. Switching between the four working modes is done automatically by self-developed software. When the instrument is operating normally, the only thing that needs to be done manually is changing the aerosol filter every 12 or 24 hours (depending on the concentration of particles in the sample air).

## 2.2 Standard gas generator for GLY and MGLY

Based on the methods described in previous studies (Washenfelder et al., 2008; Stonner et al., 2017), we designed a standard gas generator that uses high purity nitrogen (>99.999%), mass flow controller (F-201EV-MAD-22-V, 5 slm, Bronkhorst, The Netherland), U-type tube, cold trap, on-off valves, three-way valves, mix chamber, pump (50358, Thomas, Gardner Denver, Germany), and pure GLY or MGLY powder, as shown in Fig. 3. The monomeric GLY and MGLY purification methods are the same as those described in the previous literature (Washenfelder et al., 2008; Pang et al., 2014). The

operation of the standard gas generator is divided into the following three steps: (1) Passing high purity nitrogen over monomeric GLY or MGLY in the cold trap and transporting gaseous GLY or MGLY into the mix chamber; the temperature of cold trap (-72 °C) is achieved by mixing dry ice and ethanol. (2) Rotating the three way valves ($V_1$ and $V_2$) to make nitrogen enter the mix chamber directly to dilute gaseous GLY or MGLY. (3) Rotating the three way valves ($V_3$ and $V_4$) and opening valve ($V_6$) to connect the inlet and outlet of the mix chamber to the both ends of the air pump, which evenly mixes

the GLY or MGLY with nitrogen. While the air pump material may absorb some GLY or MGLY, it does not affect the gas mixture in any way that will impair the subsequent experiments.

Compared to the methods that produce GLY or MGLY by either heating GLY trimer dihydrate powder or MGLY powder (Gen et al., 2018) or using a temperature-controlled Teflon bubbler filled with solution (Min et al., 2016; Zarzana et al., 2018), the standard gas produced by our generator is more stable and can be maintained at a relatively constant concentration;

the 50 L mix chamber used in this study can provide a constant concentration of GLY or MGLY for approximately 20 min. Furthermore, our generator can produce GLY or MGLY concentrations on the order of ppt to ppm by adjusting the flow rate of the elution gas and the dilution ratio of the gas in the mix chamber. Results from several instrumental performance tests are recorded in Table 1. The concentrations of the fourth GLY test and second MGLY test are 0.93 ppb and 0.61 ppb,





respectively, which are close to their concentrations in ambient air (Li et al., 2014; Shen et al., 2018). During 20 min of standard gas supplement, the standard deviation of each concentration test is much smaller than the uncertainty of our IBBCEAS system, which indicates that our standard gas generator can provide good stability and reliability.

### 2.3 NO₂ photolytic converter

The NO$_2$ photolytic converter is mainly comprised of a photolytic module, power control module, and temperature control module. The photolysis cell is a 60.0 mm tube (18.0 mm O.D., 13.4 mm I.D.). When the system is operating at a flow rate of 2 L/min, the residence time is about 0.25 s. The core of the photolytic module is a set of 160 small LEDs (2865 COB, FLEDA, Taiwan) with a central wavelength of 395 nm and typical irradiance of 2000 mW/cm$^2$. The instrumental stability of the photolytic module is maintained by operating the module at a constant current ($2.5 \pm 0.01$ A) and a constant temperature

($26.0 \pm 0.1$ °C), which are controlled by the power module and temperature module, respectively. The NPC was first used as a part of a NO$_2$ measuring device that has been successfully deployed in many campaigns (Tan et al., 2018), and in this study, we use the NPC to remove NO$_2$ from the sample gas. Further details about the photolytic efficiency for NO$_2$, GLY, and MGLY are given in Sects. 4.2 and 4.3, respectively.

### 3 Data analysis

### 3.1 Determination of trace gas concentrations

The extinction coefficient, α(λ), accounts for absorption, Rayleigh scattering, and Mie scattering caused by gases and particles in the cavity and can be described mathematically as following Eq. (1):

$$\alpha(\lambda) = \left( \frac{I_0(\lambda)}{I_a(\lambda)} - 1 \right) \cdot \left( \frac{1 - R(\lambda)}{d} + \sigma_{\text{Rayl}}(\lambda) \right) \cdot \frac{1}{d_{\text{Ratio}}} , \tag{1}$$

where λ is the wavelength of light, $I_0(\lambda)$ is the reference spectrum, $I_a(\lambda)$ is the sample spectrum, $d$ is the cavity length, $R(\lambda)$

is the mirror reflectivity, $\sigma_{\text{Rayl}}(\lambda)$ is the extinction due to Rayleigh scattering, and $d_{Ratio}$ is the ratio of effective cavity length to physical cavity length. Since particles are filtered out by the aerosol filter, Mie scattering in the cavity is negligible and α(λ) can be simplified to Eq. (2):

$$\alpha(\lambda) = \sum_i^n \sigma_i(\lambda) \cdot n_i , \tag{2}$$

where $\sigma_i(\lambda)$ and $n_i$ are the absorption cross section and number density of the $i$th gas absorber, respectively.

According to Eq (1) and (2), the effective absorption cross section of each absorber is required in order to determine the number density. Therefore, high resolution absorption cross sections were obtained from the literature – NO$_2$ (Vandaele, 2002), GLY (Volkamer et al., 2005b), MGLY (Meller et al., 1991), H$_2$O (Rothman et al., 2005), O$_4$ (Thalman and Volkamer, 2013) – and the absorption cross sections of NO$_2$, GLY, MGLY, and H$_2$O were processed with the instrument function determined by the Fraunhofer structure of a measured solar spectrum (Kurucz et al., 1984). Compared to the few Hg lines emitted around 430 nm, the more Fraunhofer lines in the wavelength range of our interest (i.e., 440 – 460 nm) can be used to





generate a wavelength-dependent instrument slit function which account for the change of spectral resolution over the CCD pixels. The absorption cross sections of NO$_2$, GLY, and MGLY before and after processing with the instrument function are shown in Fig. 4. The blue, thick solid line in this figure is the reference spectrum of our IBBCEAS system, which overlaps the absorption structures of these three absorbers.

By using the α(λ) calculated by Eq. (1) and the absorption cross sections used in Eq. (2), the number density of the absorbers can be fitted simultaneously. The main algorithm of this fitting process is nonlinear least squares with a fifth-order polynomial to account for drift in light intensity and cavity throughput. All data processing was performed by DOAS Intelligent System (DOASIS) spectral fitting software (Kraus, 2006).

### 3.2 Mirror reflectivity and effective cavity length

In order to calculate α(λ) using Eq. (1), $R(\lambda)$ and $d_{Ratio}$ must be accurately quantified. We used pure nitrogen (> 99.999%) and pure helium (> 99.999%), two gases with distinct Rayleigh scattering sections, to calibrate $R(\lambda)$ according to Eq. (3):

$$R(\lambda) = 1 - d \cdot \frac{I_{N2}(\lambda) \cdot n_{N2} \cdot \sigma_{Rayl,N2}(\lambda) - I_{He}(\lambda) \cdot n_{He} \cdot \sigma_{Rayl,He}(\lambda)}{I_{He}(\lambda) - I_{N2}(\lambda)} \qquad (3)$$

where $d$ is the cavity length, $I_{N2}(\lambda)$ is the spectrum obtained in N$_2$ mode, $I_{He}(\lambda)$ is the spectrum obtained in He mode, $\sigma_{Rayl,N2}(\lambda)$ is the Rayleigh scattering cross section of nitrogen (Sneep and Ubachs, 2005), $\sigma_{Rayl,He}(\lambda)$ is the Rayleigh

scattering cross section of helium (Rao, 1977), and $n_{N2}$ and $n_{He}$ are the number density of nitrogen and helium, respectively. The reflectivity calibration results from our IBBCEAS system during a campaign in the Yangtze River Delta (YRD) , China is shown in Fig. 5. During the YRD campaign, the $R(\lambda)$ peak was 0.99993 at 439 nm and the total uncertainty of this result is 5% because of the uncertainty in the scattering cross sections of N$_2$ (Sneep and Ubachs, 2005). After the system running continuously for 36 days, the relative change of reflectivity was 0.0005% which is much smaller than the uncertainty of the

measurement results.

Because of the continuous purge gas at both ends of the cavity during sampling, the effective length of the cavity is shorter than its physical length; thus, $d_{Ratio}$ is less than 1.0. In order to determine $d_{Ratio}$, we measured three NO$_2$ concentrations in the cavity, which were prepared from a bottled standard (10.2 ppm NO$_2$) and diluted with high purity nitrogen (> 99.999%) in the multi-gas calibrator (146i). The NO$_2$ concentrations were measured with and without purge gas and denoted as $C_{NO2}^{WP}$

and $C_{NO2}^{WTP}$, respectively. When calculating $C_{NO2}^{WP}$ and $C_{NO2}^{WTP}$ simultaneously with Eq. (1) and (2), we assume that $d_{Ratio}$ is equal to 1.0. Under this assumption, $C_{NO2}^{WP}$ is less than $C_{NO2}^{WTP}$ when the same concentration of NO$_2$ is being measured because the effective length of the cavity is overestimated. The measurement results of $C_{NO2}^{WP}$ and $C_{NO2}^{WTP}$ are shown in Fig. 6, which illustrates that the $C_{NO2}^{WP}/C_{NO2}^{WTP}$ ratio is constant at 0.89 (blue line in Fig. 6) for the three NO$_2$ concentrations. The only variable in Eq. (1) and (2) is $d$; therefore, the $C_{NO2}^{WP}/C_{NO2}^{WTP}$ ratio is equal to the $d_{Ratio}$. Because of the 4% uncertainty of the NO$_2$

absorption cross section (Vandaele, 2002) and 2% uncertainty of the NO$_2$ prepared by the 146i, the uncertainty of the $d_{Ratio}$ is approximately 4%.



## 4 Results and discussion

### 4.1 Instrument performance

#### 4.1.1 Limit of detection and uncertainty

The precision of the IBBCEAS system can be estimated by calculating the Allan deviation (Allan, 1966), which is an

algorithm that is commonly used for optical cavity technology (Brown et al., 2002; Langridge et al., 2008; Duan et al., 2018). We continuously measured 13600 spectra in Reference mode over 4 hours and designated the average value of the first 100 spectra as $I_0$. The remaining 13500 spectra were averaged in sets of 2, 4, 6, …, and ultimately 6750. Since the integration time of each spectrum is 1.0 s, we obtained a time series, $I_a$, that contains 6750 spectra with 2.0 s integration time, 3375 spectra with 4.0 s integration time, and so forth up to 2 spectra with 6750 s integration time. A time series of α at 439 nm

was calculated using Eq. (1) and its Allan variance was determined using Eq. (4):

$$\sigma_{A_\alpha}^2(t) = \frac{1}{2(n-1)} \sum_{i=1}^{n-1} [\alpha_{i+1}(t) - \alpha_i(t)]^2, \tag{4}$$

where $t$ represents the integration time, $n$ represents the number of time series, $\alpha_i(t)$ is the extinction coefficient at each integration time from $i = 1$ to $i = n$. The arithmetic square root of the $\sigma_{A_\alpha}^2(t)$, the Allan deviation $\sigma_{A_\alpha}(t)$, can be used to evaluate the instrumental precision.

The results plotted in Fig. 7 illustrate that when the integration time is less than 100 s, the sensitivity of our instrument can be improved by increasing the integration time. The instrument has the best precision when the integration time is near 100 s, after which the Allan deviation increases with integration time because of the drift of the light source. When the integration time is 100 s, the limit of detection (LOD) can be estimated by calculating the standard deviation of each absorber concentration retrieved from the 135 averaged spectra. The LOD of our IBBCEAS system in 100 s is estimated to be 18 ppt

for $NO_2$, 30 ppt for GLY, and 100 ppt for MGLY.

According to Gaussian error propagation, the uncertainty associated with measurements of gas absorbers can be estimated using the uncertainty of the absorber's $\sigma(\lambda)$, $R(\lambda)$, $d_{Ratio}$, temperature, and pressure. For our IBBCEAS system, the uncertainty of $R(\lambda)$ is 5%, which is dominated by the uncertainty of the scattering cross sections of $N_2$. The uncertainty of the $d_{Ratio}$ is approximately 4.5% and those of temperature and pressure are each 0.5 %. The uncertainties of $NO_2$, GLY, and

MGLY can be found in the literature and are 4% (Vandaele, 2002), 5% (Volkamer et al., 2005b), and 15% (Meller et al., 1991), respectively. Based on the above parameters, the accuracy of our IBBCEAS system is estimated to be ± 8 % for $NO_2$, ± 8 % for GLY, and ± 16 % for MGLY.

### 4.1.2 Sample loss

Based on the standard gas generator for GLY and MGLY described in Sect. 2.2, experiments investigating sample loss in the

sampling line were performed as follows. First, four Teflon sampling lines with length equal to 1 m, 3 m, 5 m, and 7 m were prepared. Second, the IBBCEAS system and standard gas generator were connected using the 1 m sampling line and gas was



pumped into the IBBCEAS system for measurement. Third, replace the sampling line every four minutes in the following order: 3 m, 5 m, 7 m, and 1 m. The experiment was done twice for each length of sampling line and the results are shown in Fig. 8a. The concentrations measured during the first set of experiments fluctuated near 1.06 ppb and the concentrations measured during the second set fluctuated near 0.60 ppb. The two sets of experimental results demonstrated that sample loss

is negligible in sampling line when its length is less than 7 m. Similarly, experiments investigating sample loss in the filters was performed using four filters with different levels of cleanliness (see Fig. 8c). Filter #1 is a clean filter that has never been used and #2, #3, and #4 are used filters that were saved during field campaigns; the daily average concentrations of PM2.5 corresponding to these filters are 11 $\mu g/m^3$, 37 $\mu g/m^3$, and 83 $\mu g/m^3$, respectively. During the sample loss experiments, the length of sampling line between the IBBCEAS system and the standard gas generator was fixed at 1 m and change the filter

at the end of the sampling line in the order of #1, #2, #3, #4, and #1. The concentration of GLY was constant, $0.55 \pm 0.02$ ppb, when using the different filters (see Fig. 8b).

The above experiments demonstrate that GLY sample loss is negligible in both the sampling line and filter of our IBBCEAS system, which is consistent with the results from previous studies (Washenfelder et al., 2008; Min et al., 2016). The results from previous studies indicate that MGLY is even less reactive than GLY as the effective Henry's law consistent of MGLY

is much smaller than that of GLY (Betterton and Hoffmann, 1988) and the gas-particle partitioning constant for MGLY is at least 30 times lower than that of GLY (Kroll et al., 2005); therefore, any loss of MGLY to the sampling line and filter should also be negligible.

### 4.1.3 Interference of NO₂ on spectra fitting

An example of spectra fit for one measurement during the YRD campaign is shown in Fig. 9. The air pollution events at this

rural site are mostly dominated by biomass burning, so relatively high concentrations of GLY and MGLY were measured. The wavelength range that we chose for quantifying GLY and MGLY includes strong structured absorption of NO₂. Furthermore, the concentration of NO₂ in the troposphere is much higher than that of either GLY or MGLY, especially during the air pollution events in China; therefore, the presence of high NO₂ concentrations may affect the spectral fitting of GLY and MGLY. In order to verify this conjecture, we processed the data obtained during two campaigns as follows: we

plotted the changes in NO₂ concentration and spectra fitting residual over time on the same graph to check whether they have the same general trend, normalized the NO₂ concentration and fitting residual, and performed correlation analysis. Figure 10a illustrates that the NO₂ concentration and fitting residual trends of the processed data from the YRD campaign are similar, especially when the mixing ratio of NO₂ is greater than 10 ppb. The fitting residual is approximately $3.0 \times 10^{-9}$ $cm^{-1}$ when the NO₂ mixing ratio is approximately 50 ppb. The correlation coefficient ($R^2$) of these two normalized

parameters is 0.949, which indicates very good agreement.

During the PKU campaign, we attached the NO₂ photolytic convertor (NPC), described in Sect. 2.3, to the front of our IBBCEAS instrument to reduce the concentration of NO₂ in the sampled gas; analysis of the experimental results are given in Fig. 10c and 10d. In addition to our IBBCEAS system, another instrument for measuring NO₂ (42i, Thermo Fisher



Scientific Inc., Waltham, MA, USA) was deployed during this campaign. Figure 10c shows that the $NO_2$ concentration measured by our IBBCEAS system is much lower than that measured by the 42i when the NPC is being used. When using the NPC, the correlation coefficient of the $NO_2$ concentration and fitting residual drops to 0.88 and the fitting residual falls to $5.0 \times 10^{-10}$ $cm^{-1}$ while the ambient $NO_2$ mixing ratio is still approximately 50 ppb. Based on the above analysis, high

concentrations of $NO_2$ interfere with the spectral fitting and this interference can be reduced by using the NPC. The $NO_2$ conversion efficiency of the NPC and its effect on the measured GLY and MGLY concentrations will be discussed in the following sections.

**4.2 GLY measurements**

In order to determine the $NO_2$ removal efficiency of our NPC, we prepared a concentration gradient of $NO_2$ gas samples,

which were produced from a bottled standard (10.2 ppm $NO_2$) and diluted with high purity nitrogen (> 99.999%) in the multi-gas calibrator; each $NO_2$ concentration was measured twice, with the NPC on and off. The measurement results are shown in Fig. 11a, which illustrates that the removal efficiency of the NPC is constant at approximately 76% for the different concentrations of $NO_2$. Stability tests of the instrument were also performed and indicate that the efficiency does not change over time on the scale of hours. The impact of the NPC on the measured GLY concentration was tested using a similar

method wherein the constant concentration of GLY produced by the standard gas generator was the gas to be measured. Based on the results shown in Fig. 11b, the NPC also photolyzes a small fraction of the GLY (approximately 5%). Therefore, when the NPC is working, the GLY concentration obtained by spectra fitting needs to be corrected by dividing by 95%.

When repeating the above experiments using well-mixed $NO_2$ and GLY as the gas to be measured, we observed an interesting phenomenon whereby the concentration of $NO_2$ dropped rapidly while the NPC was running and the

concentration of GLY increased. After the NPC was turned off, the concentrations of the two compounds returned to the same levels as before the NPC was turned on (see Fig. 11c). We conducted another experiment to prove that this phenomenon was not accidental. First, we prepared standard GLY and $NO_2$ gases and stored them in separate PTFE bags. Second, we mixed the GLY and $NO_2$ standard and delivered it to the IBBCEAS system. Third, we fixed the concentration of GLY in the cavity and gradually reduced the concentration of $NO_2$. Based on the spectra fitting results (see Fig. 12a), the

concentration of GLY increased as that of $NO_2$ decreased, although we manually reduced the $NO_2$ concentration without changing that of GLY. Therefore, there seems to be a competitive relationship between the spectra fitting of $NO_2$ and GLY.

In order to further verify the observed phenomenon, we attempted to generate spectra to simulate the experimental $NO_2$ and GLY gas tests. The spectra were created by the following steps: (1) Set the $NO_2$ concentration $n_{NO_2}$ and GLY concentration $n_{GLY}$ to the value to be studied and substitute $n_{NO_2}$, $\sigma_{NO2}(\lambda)$, $n_{GLY}$, and $\sigma_{GLY}(\lambda)$ into Eq. (2) to calculate $\alpha(\lambda)$, where $\sigma_{NO2}(\lambda)$

and $\sigma_{GLY}(\lambda)$ are the absorption cross sections of $NO_2$ and GLY after processing with the instrument function. (2) Take the spectrum obtained during the reference mode as $I_0(\lambda)$ and substitute $I_0(\lambda)$, $R(\lambda)$, $d$, $\sigma_{Rayl}(\lambda)$, $d_{Ratio}$, and $\alpha(\lambda)$ calculated in step (1) into Eq. (1). As all the parameters except $I_a(\lambda)$ in Eq. (1) are already determined, $I_a(\lambda)$ can be yielded. (3) Add a set





of random numbers between 100 and 1000 representing the noise of the system to the intensity corresponding to each wavelength of $I_a(\lambda)$; this range, 100 and 1000 counts, was used because it is close to the actual noise level of our IBBCEAS system. We set the concentration of GLY to 1 ppb and the concentration of $NO_2$ to 0, 15, 30, 45 ppb to generate a series of $I_a(\lambda)$, and then calculated the concentration of these two gas absorbers from the generated $I_a(\lambda)$. Results from the spectral simulations are shown in Fig. 12b. It can be found that the retrieved GLY concentration is lower than its setting value while vice versa for $NO_2$. This is consistent with the experimental results discussed above, and could be caused by the lack of Rayleigh scattering and Mie scattering in these $I_a(\lambda)$ simulations or other unknown reasons. Therefore, it is obvious that the spectral resolving of $NO_2$ and GLY is competing with each other. Since $NO_2$ has a higher ambient concentration and stronger absorption structure than GLY, the GLY concentration determined by IBBCEAS could be underestimated in the presence of $NO_2$ and the higher the $NO_2$ concentration, the greater the underestimation. Because whether the random number can represent the noise of the whole system is uncertain, it is difficult to evaluate the uncertainty of the spectral simulation, this method is only suitable for qualitative analysis instead of quantitative analysis presently. Further research is required to modify or parameterize the underestimation of GLY concentration and correct the measured value to be closer to its true value in ambient air.

## 4.3 MGLY measurements

Compared to $NO_2$ and GLY, the absorption cross section of MGLY is less structured, which means it is difficult to accurately calculate its concentration using spectral fitting. As demonstrated by the results in Table. 1, the difference in the measured MGLY concentrations is greater than those of GLY although the GLY and MGLY standard gases were produced by the same method and measured using the same instrument. Hence, each step in the process of MGLY spectral fitting needs to be considered carefully.

Selecting reasonable spectral fit ranges is necessary in order to accurately fit the concentration of MGLY; we chose four spectral fit ranges based on the structure of the absorption cross section of MGLY: 440-451 nm, 445-453 nm, 440-453 nm, and 430-453 nm (see Fig. 13a). Each spectral fit range was used to fit two experimental sets of MGLY measurements and the results are shown in Fig. 13b. Although the MGLY concentrations were determined using the same experimental data, the fitting results from different spectral ranges varied greatly. The results of fit range (1) and fit range (3) are similar and the range of results of (3) is relatively smaller. Fit range (2) covers the narrowest wavelength range and the MGLY concentrations from (2) are discrete, especially when the mixing ratio of MGLY is approximately 4 ppb. In contrast, fit range (4) covers the widest wavelength range and its fitting results are not ideal enough, even accounting for negative values when MGLY concentration is low. Based on these results, we prefer to use fit range (3), which covers the wavelength range from 440 to 453 nm, to determine MGLY concentrations in our studies.

In order to study the effect of $NO_2$ on MGLY measurements, experiments similar to those described in Sect 4.2 were conducted. First, prepared MGLY standard gas was passed through the NPC and measured by the IBBCEAS instrument. The results in Fig. 14a show that the NPC has no effect on the measured concentration of MGLY, which is different from the





effect of the NPC on GLY. However, a similar phenomenon was observed when we repeated the above experiments with a mixture of MGLY and $NO_2$ (see Fig. 14b); the concentration of $NO_2$ dropped immediately once the NPC was turned on and the concentration of MGLY increased slightly, which is the same phenomenon that was observed for GLY. These results suggest that the MGLY concentration determined by IBBCEAS could also be underestimated in the presence of $NO_2$.

Spectral simulations were also performed to investigate the accuracy of the measured MGLY concentrations in the presence of $NO_2$. We set the concentration of MGLY to 1 ppb and the concentration of $NO_2$ to 0, 5, 10, 15, 20, 25 ppb in order to generate a series of $I_a(\lambda)$ using the same algorithm as in Sect. 4.2. The results of the simulations are shown in Fig. 16c, which illustrates that as the concentration of $NO_2$ increases, the concentration of MGLY will be underestimated. Therefore, in the presence of high $NO_2$ concentrations, measured MGLY concentrations may be lower than the real concentrations.

## 10 4.4 Comparisons to existing instruments

Comparisons of our IBBCEAS system with other instruments which are able to simultaneously measure GLY and MGLY within a time resolution of 30 minutes were made in Table. 2. For the GLY measurement, ACES, PFBHA-GC-MS, LIP, and CE-DOAS are available with detection limit values of 11-75 ppt and for the MGLY measurement, PTR-ToF-MS, CE-DOAS, PFBHA-GC-MS are available with detection limit values of 22-185 ppt. Compared with the existing instruments, the ability

of our IBBCEAS to detect GLY and MGLY is comparable. From a comprehensive perspective, the new IBBCEAS has a good performance and can be used to simultaneously measure the concentration of GLY and MGLY in the atmosphere. Compared with the ACES developed by Min et al., which is also be used to measure GLY and MGLY (Min et al., 2016), under the same integration time (100 s), the Allan deviation of their system (about $1.5 \times 10^{-10}$ cm$^{-1}$) is higher than that of our system ($8.4 \times 10^{-11}$ cm$^{-1}$), which indicates that the IBBCEAS developed by us has a better instrumental precision and

stability. With respect to the measurement interference of $NO_2$, existing CEAS do not have a good method to solve the problem. For the IBBCEAS used by Thalman et al., the systematic bias was characterized as ~ 1ppt GLY/ppb $NO_2$, and ~ 5 ppt MGLY/ppb $NO_2$. At low $NO_2$ concentration (below 10 ppb), the small effect on GLY and MGLY retrievals was unnoticeable (Thalman et al., 2015). In contrast, before entering the cavity of our IBBCEAS, the $NO_2$ in sampled air is reduced by 76 %, so the systematic bias of our system caused by $NO_2$ can be reduced accordingly. Furthermore, during

severe air pollution events, the $NO_2$ concentration in the optical cavity was always controlled between 10 and 20 ppb (see Fig. 10c), which further ensures the minimization of $NO_2$ interference on measurements of GLY and MGLY.

## 5 Conclusions

We have developed and characterized an IBBCEAS instrument for simultaneously measuring $NO_2$, GLY, and MGLY in ambient air. Based on self-developed software, the entire system is highly automated; the only thing that needs to be done

manually is replacing the particle filters during normal operation in field campaigns. Because of the uncertainties in the absorption cross sections, effective cavity length, and mirror reflectivity, the accuracies of the measured concentrations are




estimated to be ± 8% for NO$_2$, ± 8% for GLY, and ± 16% for MGLY. Compared to IBBCEAS systems for the measurement of GLY and MGLY discussed in the existing literature, our novelties are mainly reflected in the following:

(1) A standard gas generator has been set up to provide a constant concentration of GLY or MGLY that can be maintained down to approximately 200 ppt, which is similar to their real concentrations in troposphere. The standard gas generator enables systematic experiments investigating sample loss and characterizing the IBBCEAS system.

(2) The interference of high NO$_2$ concentration on spectra fitting, and subsequently determining the concentrations of GLY and MGLY, is analyzed and discussed using both measured results and spectral simulations. In order to minimize the effect of NO$_2$ on GLY and MGLY, a NO$_2$ photolytic converter was used to remove NO$_2$ in the sampled air.

In summary, sample loss experiments with our IBBCEAS system demonstrated that sample loss of GLY and MGLY in the sampling line and particle filter are negligible. In terms of the interference of NO$_2$ on GLY and MGLY measurements, the spectral fit residual increases as the NO$_2$ concentration increases when all other conditions are the same. Furthermore, the measured GLY and MGLY may be underestimated in the presence of high NO$_2$ concentrations. By utilizing the NPC to remove sampled NO$_2$, the spectral fit residual is effectively reduced and the measured GLY and MGLY concentrations will be more accurate, such that the measured concentrations will be closer to their actual concentrations.

In order to accurately measure GLY and MGLY, the following methods could be developed to reduce the interference from NO$_2$. First, the sampled gas could be pre-treated to reduce the NO$_2$ concentration as much as possible without affecting GLY and MGLY. As discussed above, the higher NO$_2$ concentration, the greater underestimation of GLY and MGLY concentration, so reducing the NO$_2$ concentration can improve the accuracy of GLY and MGLY measurement results. The second method would be quantifying the competitive relationships in spectra fitting between NO$_2$ and both GLY and MGLY through laboratory experiments and theoretical calculations. After simultaneous retrieving concentrations of NO$_2$, GLY, and MGLY, concentrations of GLY and MGLY could be corrected using the parametric relationship; however, because of the complexity of the actual atmosphere, parametric results obtained in the laboratory may not be able to be extended to field campaigns. The third option could be to develop a suitable method for removing only GLY and MGLY in sampled air and regard it as a new reference mode. By making the system switch between new reference mode and sample mode frequently, the spectra acquired in both modes will include the absorption of NO$_2$ and the spectra fitting will no longer be affected by NO$_2$. Unfortunately, such methods with sufficient specificity to selectively remove GLY and MGLY are not currently available. Moreover, the iterative algorithm used in CE-DOAS (Horbanski et al., 2019) could be helpful to accurately measure the concentration of the three at the same time.

**Data availability.**

The datasets used in this study are available from the corresponding author upon request (li_xin@pku.edu.cn).



**Competing interests.**

The authors declare that they have no conflict of interest.

**Acknowledgements**

This work was supported by the National Key R&D Program of China (2017YFC0209400) and by the National Natural
Science Foundation of China (91644108). We thank Dr. Qingyu Liu and Ms. Xuewei Lu from Institute of Chemistry,
Chinese Academy of Science for their help on purifying GLY and MGLY solid standards.

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



**Table 1.** Measurements of gaseous GLY and MGLY prepared by our standard gas generator and supplied for 20 min.

| Species | Average concentration (ppb) | Standard deviation (ppb) |
| --- | --- | --- |
| GLY | 9.36 | 0.05 |
| GLY | 5.73 | 0.03 |
| GLY | 1.68 | 0.03 |
| GLY | 0.93 | 0.03 |
| MGLY | 3.99 | 0.10 |
| MGLY | 0.61 | 0.08 |



**Table 2.** Comparisons of the new IBBCEAS system with other instruments.

| Instrument | GLY Detection Limit (ppt) | MGLY Detection Limit (ppt) | Time resolution | References |
|---|---|---|---|---|
| IBBCEAS | 18 | 100 | 100 s | This work |
| ACES | 34 | Not mentioned | 5s | Min et al., 2016 |
| PFBHA-GC-MS | 75 | 185 | 30 min | Pang et al., 2014 |
| PTR-ToF-MS | 250 | 22 | 10 min | (Stonner et al., 2017; Michoud et al., 2018) |
| LIP | 11 | 243 | 5 min | Henry et al., 2012 |
| CE-DOAS | 19 | 170 | 1 min | Thalman and Volkamer, 2010 |



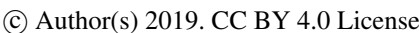



**Figure 1. (a)** Optical cavity module of the IBBCEAS system. Schematic layout of the instrumental flow system depicting four working modes: **(b)** $N_2$ mode; **(c)** He mode; **(d)** Reference mode; **(e)** Sample mode.




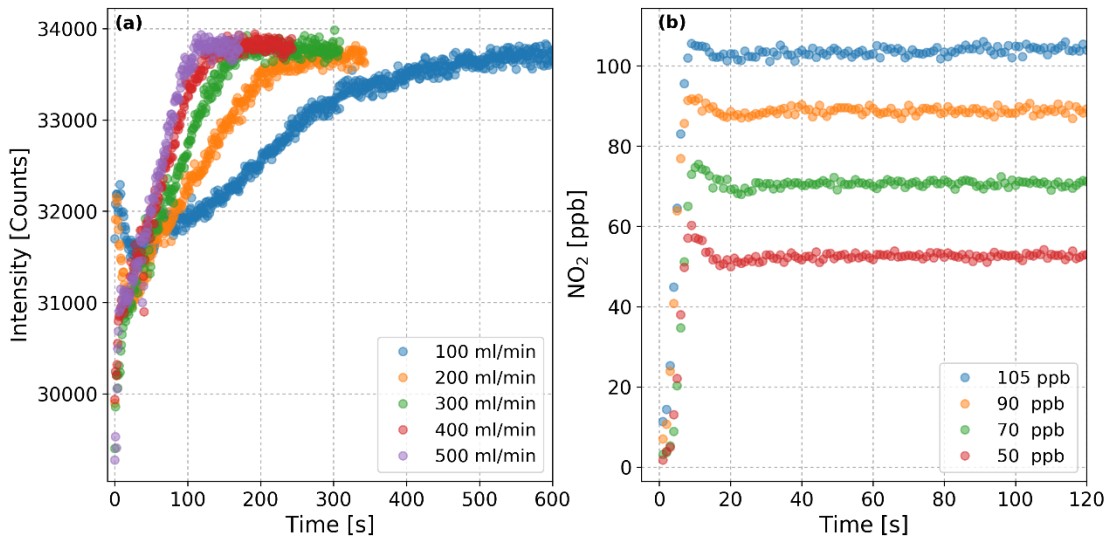

**Figure 2.** The time required to achieve a stable signal when switching between different working modes: **(a)** changes in intensity at 450 nm when switching from $N_2$ mode to He mode at different He flow rates; **(b)** changes in $NO_2$ concentration when switching from Reference mode to Sample mode with different $NO_2$ sample concentrations.



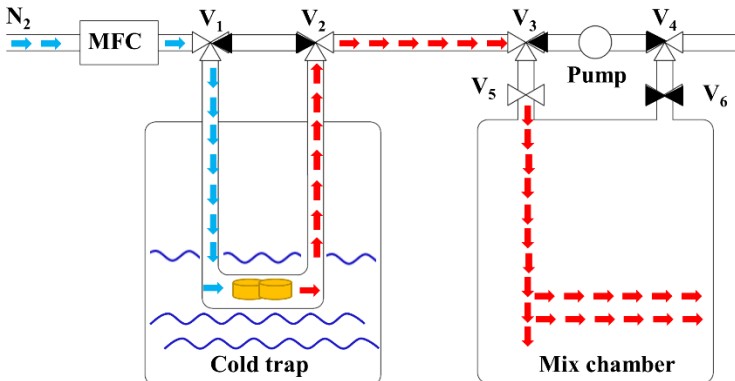

**Figure 3.** Diagram of the standard gas generator for GLY and MGLY.

10.5194/amt-2019-40
Atmospheric Measurement Techniques




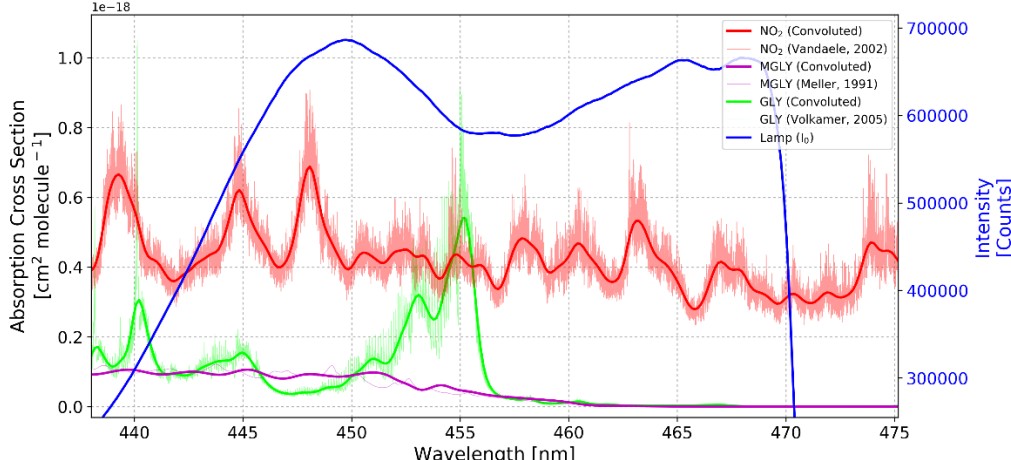

**Figure 4.** Reference spectrum of the IBBCEAS system (blue) and absorption cross sections of NO₂, GLY, and MGLY. The thin lines are the high resolution cross sections documented in the literature and the thick lines are the cross sections after processing with the instrument function determined by the Fraunhofer reference spectrum (Kurucz et al., 1984).





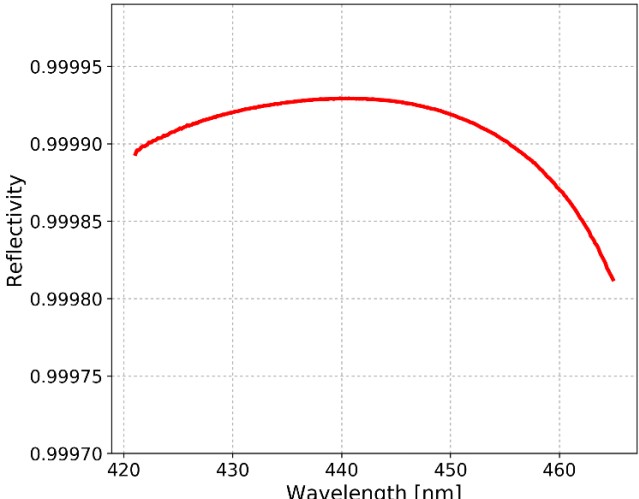

**Figure 5.** Mirror reflectivity, $R(\lambda)$, during the YRD campaign.



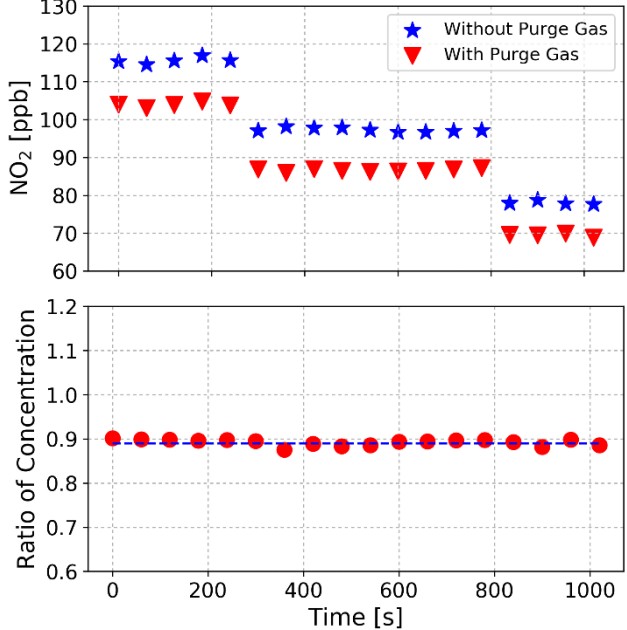

**Figure 6.** NO$_2$ concentrations measured with and without purge gas. The ratio of NO$_2$ concentration under different conditions is nearly constant at 0.89 (blue dotted line).




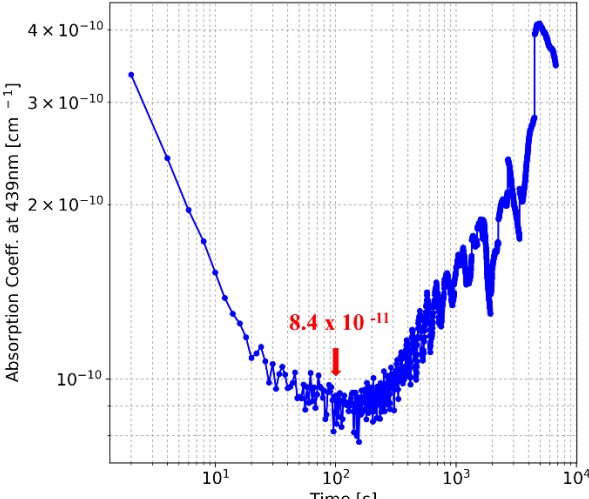

**Figure 7.** Allan deviation at 439 nm. The precision ($\mathbf{1\sigma}$) of the instrument is $\mathbf{8.4 \times 10^{-11}\ cm^{-1}}$ for an integration time of 100 s.

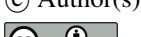



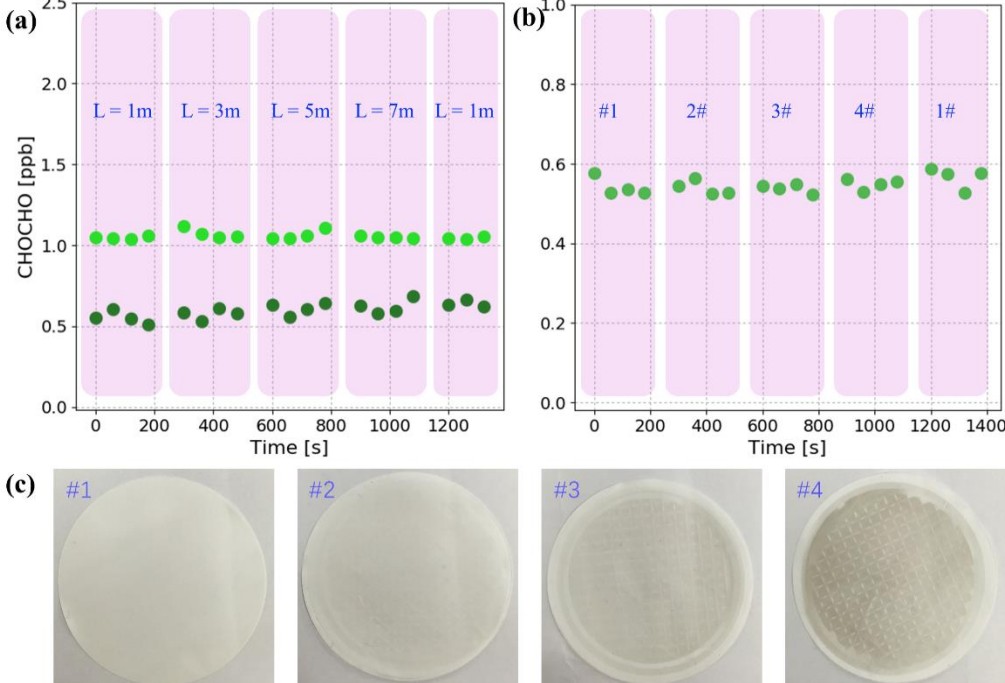

**Figure 8.** Sample loss experiments: **(a)** GLY concentrations measured with different lengths of sampling line; **(b)** GLY concentrations measured using particle filters with different levels of cleanliness. **(c)** Samples of the four particle filters corresponding to **(b)**; the daily average concentrations of PM2.5 corresponding to these filters are 0 **µg/m³** (new filter), 11 **µg/m³**, 37 **µg/m³** and 83 **µg/m³**, respectively.





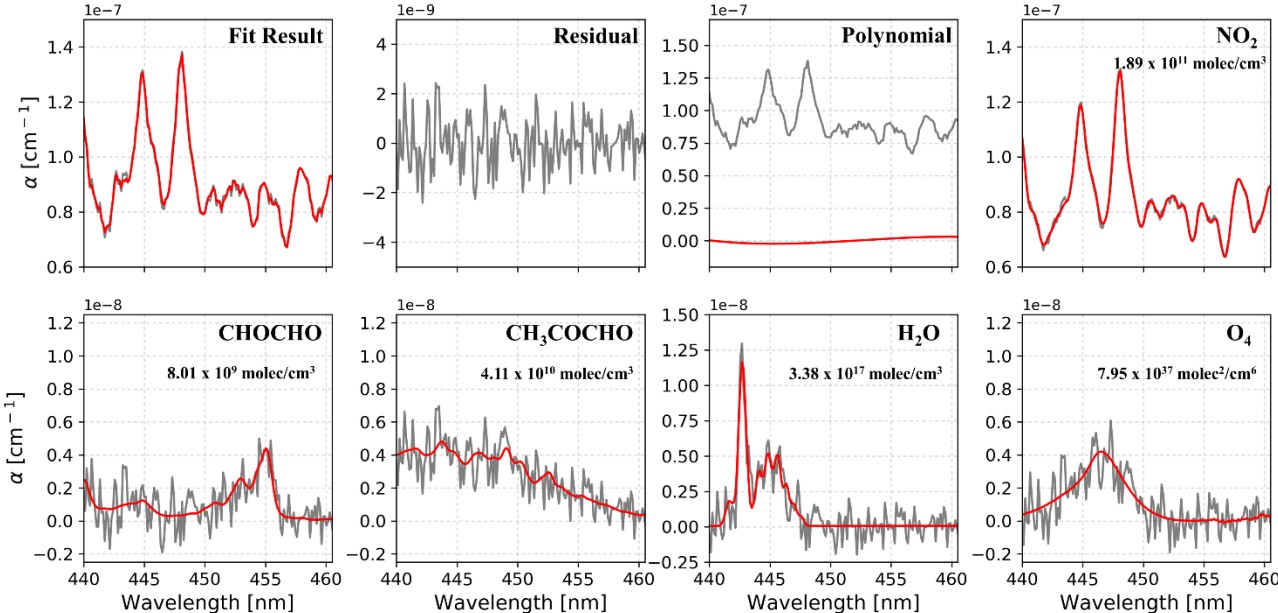

**Figure 9.** An example of spectral fit for one measurement during the YRD campaign.



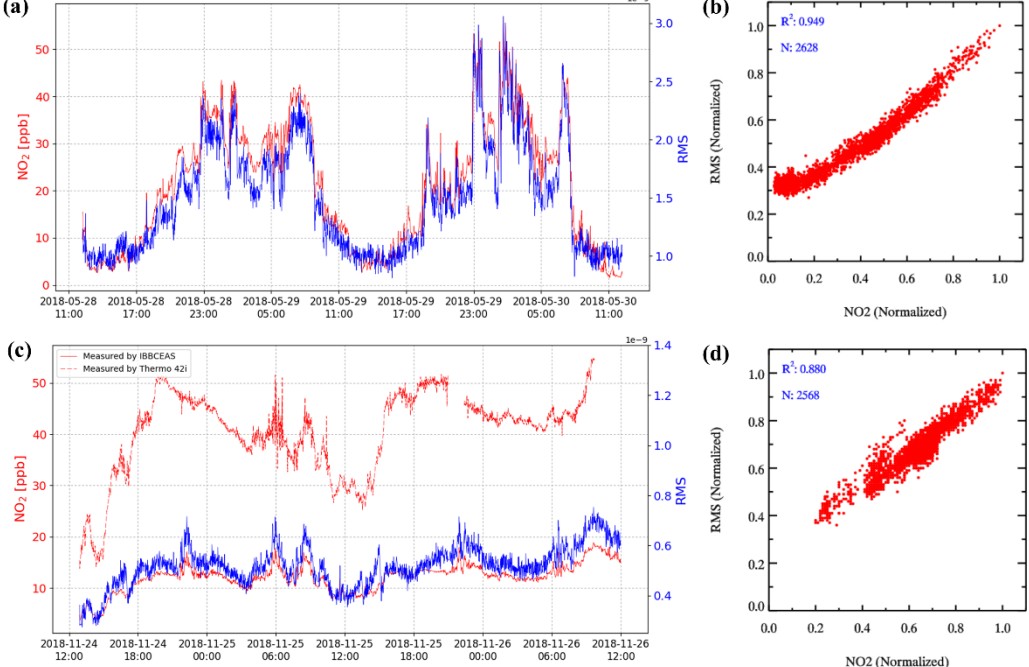

**Figure 10. (a)** Time series and **(b)** correlation plot of NO$_2$ concentration and fitting residual from the YRD campaign. **(c)** Time series and **(d)** correlation plot of NO$_2$ concentration and fitting residual from the PKU campaign.

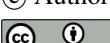


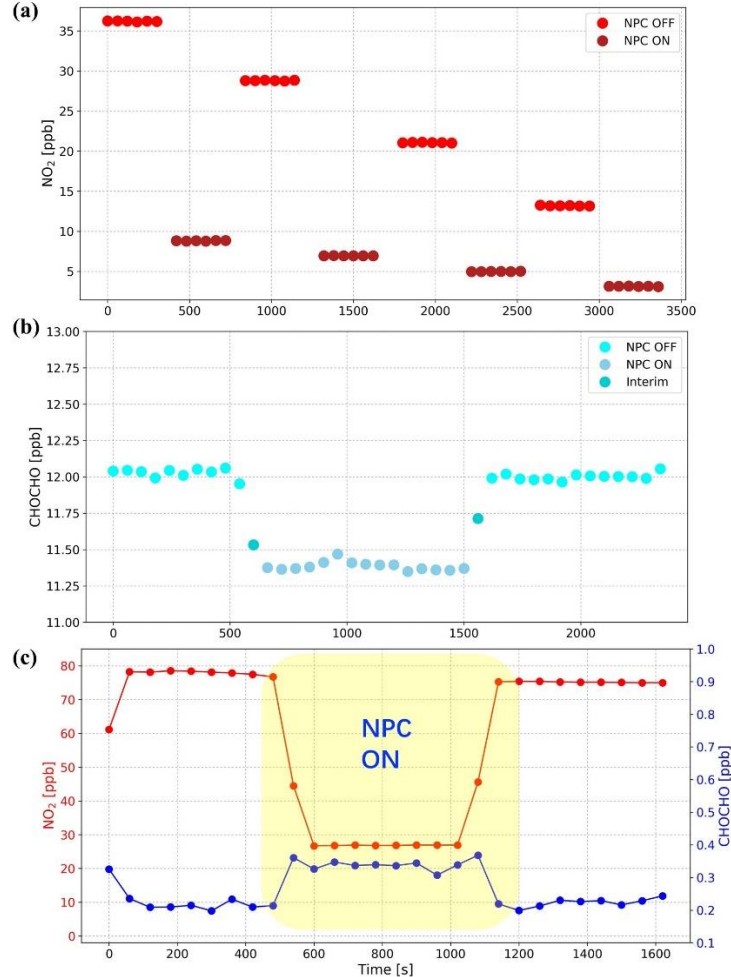

**Figure 11.** Results of photolysis experiments: **(a)** NO₂, **(b)** GLY, **(c)** well-mix NO₂ and GLY.





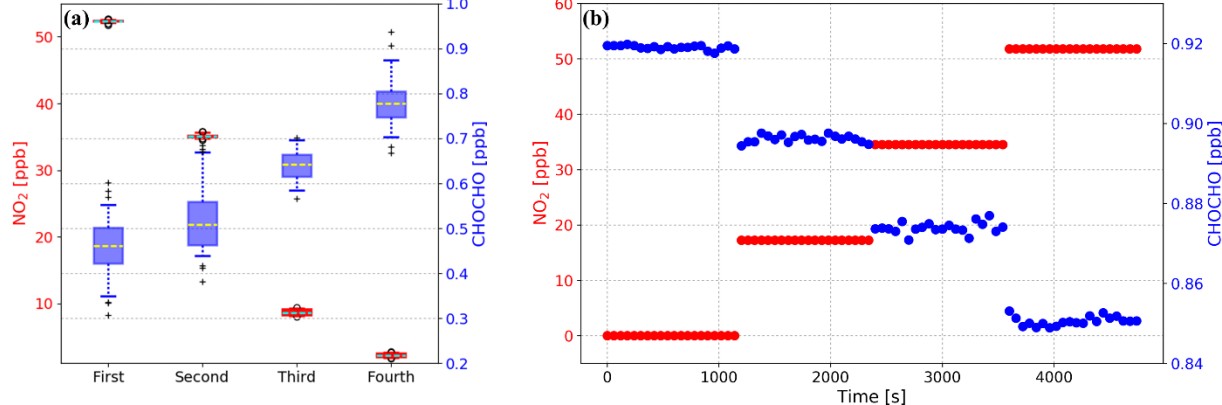

**Figure 12.** Impact of fixing the concentration of GLY and changing that of $NO_2$ on spectral fitting: **(a)** measurement results from the laboratory experiment; **(b)** results from the simulation experiment.





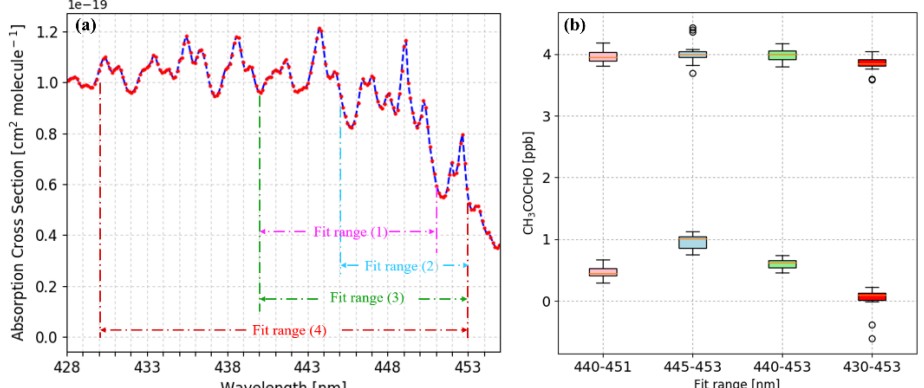

**Figure 13.** (a) Four spectral fit ranges for MGLY and (b) the corresponding concentrations of MGLY.





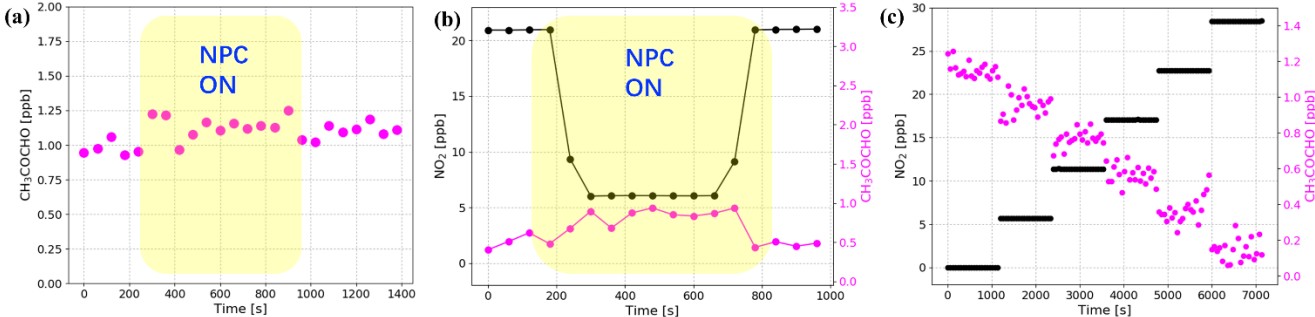

**Figure 14. (a)** Photolysis experiment of MGLY; **(b)** photolysis experiment of a MGLY and NO$_2$ mixture; **(c)** results from simulated spectra of a NO$_2$ and MGLY mixture.

