# Peer review of "An IBBCEAS system for atmospheric measurements of glyoxal and methylglyoxal in the presence of high NO2 concentrations"

_Atmospheric Measurement Techniques, 2019_

## Referee Comment (RC1) · Anonymous Referee #3 · 2 Apr 2019

General comments

This paper reported the development of an incoherent broadband cavity enhanced absorption spectrometer for simultaneous measurement of NO2, glyoxal (GLY) and methylglyoxal (MGLY). A NO2 photolytic convertor (NPC) was used to minimize the interference of high concentration NO2 to GLY. The photolysis of NO2 can lead to the formation of O3. My major comment is that if the photolysis of ambient air can potentially generate artificial GLY or MGLY, especially in VOCs rich environments.

Specific comments

1, page 2, line 17. A careful survey of GLY instruments is encouraged. A short discussion about recent IBBCEAS systems used for GLY measurements and their detection limits is better than only a sentence of other gas's applications.

2, page 2, line 28. I note a paper recently accepted for publication in AMT that also tried to fix the problem of NO2 interference to GLY, which should be included in the introduction.

Liang, S., Qin, M., Xie, P., Duan, J., Fang, W., He, Y., Xu, J., Tang, K., Meng, F., Ye, K., Liu, J., and Liu, W.: Development of an incoherent broadband cavity-enhanced absorption spectrometer for measurements of ambient glyoxal and NO2 in a polluted urban environment, Atmos. Meas. Tech. Discuss., https://doi.org/10.5194/amt-2018-430, in review, 2018.

3, page 5. How to determine the absolute GLY concentration in this work is still not clear, by measuring the pressure or flow rate?

4, page 7, line 19. The relative change of 1-R is more meaningful than R.

5, page 11, sec. 4.2. Please include the fitting residual information and give some discussion about the "unknown reasons". Did the authors shift or stretch the reference spectrum?

6, page 12, sec. 4.4. Please include the comparisons with other IBBCEAS systems. The sensitivity of Min et al.'s was $1.5 \times 10^{-10}$ cm-1, and the authors' was $8.4 \times 10^{-11}$ cm-1 (with 100 s integration time, line 18 in the text). Table 2 is not clear (5s, 100s). The corresponding time for each detection limit needs to be clearly stated. Furthermore, please carefully check if the data used in Table 2 are correct.

7, page 23, Fig. 4, please check the convolution of MGL reference is correct. There is an obvious shift, and the peaks are vanished.

8, page 27, Fig. 8(a). The symbols are not clearly indicated.

---

## Referee Comment (RC2) · Anonymous Referee #1 · 3 Apr 2019

Liu et al. describe a new incoherent broadband cavity enhanced absorption spectrometer (IBBCEAS) for atmospheric measurements of NO2, glyoxal, and methyl glyoxal. The instrument has been thoroughly characterized and performs well compared to existing methods. As a new feature, the instrument used a photolytic converter (NPC) to selectively reduce the amount of NO2 in their sample stream.

Major revisions will be required before this paper is acceptable for publication.

(1) There are too many figures in the current draft. Many are not necessary.

(2) Some figures are unclear as the captions are too short.

(3) I was not convinced of the merits of the NPC. Is the NPC even necessary? It seems that (valid) data could be obtained without it, though at a slightly higher noise level.

(4) Does the NPC generate glyoxal and methyl glyoxal when ambient air containing organic compounds is sampled? The MCM lists 127 precursors for glyoxal and 143 precursors for methyl glyoxal. I havn't gone through these lists in detail, but it seems that more characterization experiments are warranted in my opinion. It seems to me that the NPC may generate more issues than it solves.

(5) The convolved spectrum of MGLY (Figure 4, thick purple line) does not look correct.

Specific comments

pg 2, line 28 "Thalman et al. first encountered the challenge of fitting GLY and MGLY absorption spectra in the presence of high NO2 concentrations (Thalman et al., 2015). To our knowledge, an effective method has not yet to solve this problem." This is a bit of mischaracterization - Thalman et al. [2015] stated that "For both CE-DOAS and BBCEAS ... we do not find significant bias; i.e., an upper-limit change in glyoxal due to NO2 is derived as +/-200 pptv glyoxal in the presence of 200 ppbv NO2 (or bias of 1 pptv glyoxal/1 ppbv NO2)."- see also major comment #3.

pg 7 line 31. Consider adding a digit to each uncertainty as otherwise 4% + 2% = 4%.

pg 9 line 23. "The presence of high NO2 concentrations may affect the spectral fitting of GLY and MGLY". Perhaps, but this is not obvious from Figure 9 (which seem OK to me). To make their point, it seems like the authors could perform a rather easy experiment: Deliver a constant concentration of GLY (or MGLY) and add increasing amounts of NO2 to this blend. Are the correct GLY (or MGLY) concentrations retrieved?

Figure 1 (a) Consider adding dimensions to this sketch, such as the inter-mirror distance

Figure 2 is not necessary in my opinion. Simply state the result in the text.

Figure 3. You may want to state the dimensions of the mixing chamber, temperature of the cold trap, and flow rates.

Figure 4 The convolved methylglyoxal spectrum does not look right.

Figure 6. This factor will be flow rate dependent, and the sample flow rate should be stated in the caption. Please zoom in on the ratio more (0.85 to 0.95) and state the standard deviation of this measurement. The effective path was determined from a comparison of measured to known concentration; the figure is misleading as it is suggested that it was determined by turning the purge gases on and off (filling the purge flow volumes would take much more time than shown here). Consider clarifying this in the text.

Figure 11 The figure caption should describe the experiment conducted here in more detail. Figure 11 (c) The figure caption should indicate what is meant by the yellow overlay. Are the times displayed correct? It seems that the NO2 mixing ratios stay at the high levels well after the NPC was turned on, but increase again before the NPC is turned off?

Figure 12 - what does 'fixing' the concentration mean? What is 'first', 'second', 'third', and 'fourth'?

Figure 13 (b). There are two sets of concentrations displayed - which one is methylglyoxal, and what is the other one?

Figure 14. The figure caption should indicate what is meant by the yellow overlay. (b) Are the times displayed correct? It seems that the NO2 mixing ratios stay at the high levels well after the NPC was turned on, but increase again before the NPC is turned off? The photo-oxidation of certain organic molecules (e.g., benzene, toluene etc.) can yield glyoxal or methylglyoxal. It looks like this is happening here. Consider evaluating the NPC while sampling an atmospheric background of those VOCs.

Fig 14 (c) The caption does not adequately describe what is shown in the figure.

[Figure]

---

## Author Comment (AC1) · 30 Jun 2019

We thank Referee #1 for his/her comments on our paper, which helped us to improve the quality of the paper. Below, we answer the reviewer's question point by point.

General comments

This paper reported the development of an incoherent broadband cavity enhanced absorption spectrometer for simultaneous measurement of $NO_2$, glyoxal (GLY) and methylglyoxal (MGLY). A $NO_2$ photolytic convertor (NPC) was used to minimize the interference of high concentration $NO_2$ to GLY. The photolysis of $NO_2$ can lead to the formation of $O_3$. My major comment is that if the photolysis of ambient air can potentially generate artificial GLY or MGLY, especially in VOCs rich environments.

**Reply:**

Given the size of the photolysis cell (i.e., a cylinder with 60 mm length and 13.4 mm inner diameter) and the sampling flow rate of 2 L/min, the residence time of the sampled air in the cell is about 0.25 s. In such a short period, the production of GLY and MGLY in the cell is negligible which can be verified by model simulations. The model includes the full MCM chemistry (version 3.3.1, http://mcm.leeds.ac.uk/MCM/ ) for all NMHCs and their oxidation products. The initial concentrations of OH, $HO_2$, $O_3$, NO, HONO, CO, $CH_4$, $C_2$-$C_{12}$ NMHCs are set to the average values obtained during a field observation campaign in 2018 in Yangtze River Delta, China (Table S1). The total OH reactivity due to NMHCs is about 3.1 $s^{-1}$. The relative humidity, temperature, and pressure are constrained by the campaign averages, i.e., 60%, 300 K, 1007.65 hPa, respectively. The photolysis frequencies are constrained by values calculated from the spectral actinic flux inside the cell (Fig. S1). The spectral actinic flux is estimated by the LED emission spectrum and by the concentration change of NO, $NO_2$, and $O_3$ when $NO_2$ standard (100 ppb) is running through the system. The initial values of $NO_2$, GLY, MGLY are set to 60 ppb, 100 ppt, and 100 ppt, respectively. Fig. S2 shows the calculated concentrations of $NO_2$, $O_3$, GLY, and MGLY for the period of 0.25 s residence time. While the $NO_2$ concentration drops from 60 ppb to 15 ppb leading to an increase of $O_3$ concentration by approximately 45 ppb, the change of GLY and MGLY concentrations is only around 1%. As shown in Fig. S3, the production rate of GLY and MGLY increases to around 5 ppb $h^{-1}$ which could only result in maximumly 0.34 ppt increase of GLY and MGLY. Even if we increase the initial NMHCs concentration in the model by a factor of 10, the maximum production of GLY and MGLY within 250 ms is less than 2 ppt which is only 2% of their initial concentration (Fig. S4). Since the NMHCs concentration in the model represents typical atmospheric condition influenced by urban air pollution, our model simulation clearly shows that the NPC can be used in the field observations for removing $NO_2$ without additional production of GLY and MGLY.

The above discussions are added in the Supplement.

Specific comments

1, page 2, line 17. A careful survey of GLY instruments is encouraged. A short discussion about recent IBBCEAS systems used for GLY measurements and their detection limits is better than only a sentence of other gas's applications.

**Reply:**

The following texts are added on page 2, line 24 in the revised manuscript.

"Washenfelder et al. were the first to use this technology to measure GLY. The precision ($1\sigma$) of their system is 29 ppt for a 1 min sampling time (Washenfelder et al., 2008). Under the same time resolution, Thalman and Volkamer reduced the detection limit ($2\sigma$) to 19 ppt for their LED-CE-DOAS (Thalman and Volkamer, 2010). The above two systems have been successfully applied to GLY measurements in field observations (Washenfelder et al., 2011; Coburn et al., 2014). From aspects of miniaturization and improving time resolution, Min et al. optimized Washenfelder et al.'s IBBCEAS for aircraft GLY measurements. The measurement precision ($2\sigma$) is 34 ppt in 5 s (Min et al., 2016). The IBBCEAS developed by Fang et al. has a precision of 28 ppt for GLY at 1 min averaging time. By applying Kalman filter to the retrieved concentrations, their measurement precision was improved to 8 ppt in 21 s (Fang et al., 2017)."

2, page 2, line 28. I note a paper recently accepted for publication in AMT that also tried to fix the problem of $NO_2$ interference to GLY, which should be included in the introduction.

Liang, S., Qin, M., Xie, P., Duan, J., Fang, W., He, Y., Xu, J., Tang, K., Meng, F., Ye, K., Liu, J., and Liu, W.: Development of an incoherent broadband cavity-enhanced absorption spectrometer for measurements of ambient glyoxal and $NO_2$ in a polluted urban environment, Atmos. Meas. Tech. Discuss., https://doi.org/10.5194/amt-2018-430, in review, 2018.

**Reply:**

The following texts are added in page 2, line 28 in the revised manuscript.

"Liang et al. thought that the interference was caused by the accuracy of the convoluted $NO_2$ absorption cross section and tried to solve this problem by measuring $NO_2$ cross section with their own spectrometer (Liang et al., 2019). In this case, the accuracy of the retrieved $NO_2$ and GLY concentrations will be dependent on the accuracy of the $NO_2$ cross section they measured."

3, page 5. How to determine the absolute GLY concentration in this work is still not clear, by measuring the pressure or flow rate?

**Reply:**

The absolution concentration of GLY is determined by fitting the reference spectra of $NO_2$, GLY, MGLY, etc. to the measured absorption coefficient, according to equations 1 and 2 in Page 5. We also measure the pressure and the temperature in the cavity of the IBBCEAS, so that the absolute concentration can be converted to mixing ratio. In this work, the absolute concentration of GLY in the $NO_2$ photolytic converter is determined by the IBBCEAS system. For checking the influence of NPC on the GLY

sampling efficiency, we only need to look at the relative change of the measured GLY concentration.

4, page 7, line 19. The relative change of 1-R is more meaningful than R.
**Reply:**
We revise the text according to your suggestion.

5, page 11, sec. 4.2. Please include the fitting residual information and give some discussion about the "unknown reasons". Did the authors shift or stretch the reference spectrum?
**Reply:**
We did apply shift and stretch on the reference spectra. The shift was limited within -1 nm to 1 nm and the stretch was limited within 0.9 to 1.1. The following texts are added in Sect. 4.2.
"…The fitting residual increased from $4 \times 10^{-10}$ to $2 \times 10^{-9}$ as the $NO_2$ concentration increased…. The uncertainty of simulation results is mainly caused by two reasons. (1) Random numbers could be not good enough to represent the actual noise of the whole system. Since the intensity of LED and the reflectivity of mirrors are not evenly distributed with the wavelength, the corresponding signal-to-noise ratios (SNR) are also different at different wavelength. As for our system, the SNR within 450-468 nm are higher than that at other wavelengths. If we only reduced the random number by 5 times within 450-468 nm and did not change that at other wavelengths, the fluctuation of the fitted GLY concentration was also reduced by 5 times. (2) The impact of Rayleigh scattering and Mie scattering are not explicitly considered during the simulation. In this case, whether polynomial should be added in the spectral fitting or not would be a problem. The retrieved GLY concentration by using a fifth-order polynomial was 20% higher than that without including polynomial. Therefore…."

6, page 12, sec. 4.4. Please include the comparisons with other IBBCEAS systems. The sensitivity of Min et al.'s was 1.5x10ˆ-10 cm-1, and the authors' was 8.4x10ˆ-11 cm- 1 (with 100 s integration time, line 18 in the text). Table 2 is not clear (5s, 100s). The corresponding time for each detection limit needs to be clearly stated. Furthermore, please carefully check if the data used in Table 2 are correct.
**Reply:**
The second paragraph in Sect. 4.4 is rewritten in the revised manuscript.
According to figure 8 in Min et al. (2016), we estimated the Allan deviation of their system for a 100 s acquisition time as $1.5 \times 10^{-10}$ $cm^{-1}$ ($2\sigma$). With respect to its GLY detect limitation, it was given in as 5 s average in Min et al.'s paper. We carefully checked the data used in Table 2 and made necessary revisions.

7, page 23, Fig. 4, please check the convolution of MGL reference is correct. There is an obvious shift, and the peaks are vanished.
**Reply:**
Thank you for pointing out this mistake. We tested two high resolution cross sections

of MGLY at the very beginning of our experiments, one is from Meller et al. (1991) and the other is from Staffelbach et al. (1995). In the manuscript, we mistook convolution results based on Staffelbach's spectrum instead of that based on Meller's spectrum. We revised Fig. 4 accordingly and checked the whole manuscript to ensure that the correct cross section is used in every part involving MGLY spectral fitting.

8, page 27, Fig. 8(a). The symbols are not clearly indicated.
**Reply:**
Revised accordingly in the manuscript.

**Reference**

Meller, R., Raber, W., Crowley, J. N., Jenkin, M. E., and Moortgat, G. K.: THE UV-VISIBLE ABSORPTION-SPECTRUM OF METHYLGLYOXAL, Journal of Photochemistry and Photobiology a-Chemistry, 62, 163-171, 1991.

Min, K. E., Washenfelder, R. A., Dubé, W. P., Langford, A. O., Edwards, P. M., Zarzana, K. J., Stutz, J., Lu, K., Rohrer, F., Zhang, Y., and Brown, S. S.: A broadband cavity enhanced absorption spectrometer for aircraft measurements of glyoxal, methylglyoxal, nitrous acid, nitrogen dioxide, and water vapor, Atmos Meas Tech, 9, 423-440, 2016.

Staffelbach, T. A., Orlando, J. J., Tyndall, G. S., and Calvert, J. G.: THE UV-VISIBLE ABSORPTION-SPECTRUM AND PHOTOLYSIS QUANTUM YIELDS OF METHYLGLYOXAL, J Geophys Res-Atmos, 100, 14189-14198, 1995.

Table S1. Initial concentration of species included in the model simulation.

| Species | Concentration | Species | Concentration |
|---------|---------------|---------|---------------|
| OH | 107 cm-3 | NO | 0.60 ppb |
| HO2 | 109 cm-3 | HONO | 0.45 ppb |
| CH4 | 1.9 ppm | CO | 0.33 ppm |
| O3 | 70.00 ppb | SO2 | 1.33 ppb |
| NO2 | 60 ppb | MGLY | 100.0 ppt |
| GLY | 100.0 ppt | TOLUENE | 0.515 ppb |
| C2H2 | 1.000 ppb | NC8H18 | 0.034 ppb |
| CBUT2ENE | 0.050 ppb | EBENZ | 0.140 ppb |
| C2H4 | 0.900 ppb | MXYL | 0.045 ppb |
| C2H6 | 2.130 ppb | NC9H20 | 0.019 ppb |
| IC4H10 | 0.380 ppb | OXYL | 0.076 ppb |
| IC5H12 | 0.330 ppb | STYRENE | 0.017 ppb |
| NC4H10 | 0.650 ppb | IPBENZ | 0.012 ppb |
| NC5H12 | 0.240 ppb | PBENZ | 0.014 ppb |
| PENT1ENE | 0.004 ppb | METHTOL | 0.016 ppb |
| TPENT2ENE | 0.002 ppb | PETHTOL | 0.015 ppb |
| C5H8 | 0.233 ppb | NC10H22 | 0.017 ppb |
| CPENT2ENE | 0.003 ppb | TM135B | 0.014 ppb |
| M22C4 | 0.019 ppb | OETHTOL | 0.036 ppb |
| M23C4 | 0.020 ppb | TM123B | 0.014 ppb |
| M2PE | 0.100 ppb | PXYL | 0.045 ppb |
| M3PE | 0.076 ppb | NC11H24 | 0.019 ppb |
| HEX1ENE | 0.009 ppb | C3H8 | 2.010 ppb |
| NC6H14 | 0.130 ppb | C3H6 | 0.120 ppb |
| M2HEX | 0.030 ppb | C4H6 | 0.005 ppb |
| CHEX | 0.044 ppb | TM124B | 0.016 ppb |
| M3HEX | 0.041 ppb | TBUT2ENE | 0.002 ppb |
| BENZENE | 0.364 ppb | BUT1ENE | 0.040 ppb |
| NC7H16 | 0.055 ppb | | |

[Figure]

Figure S1. Spectral actinic flux inside the photolysis cell of the NO$_2$ convertor.

[Figure]

Figure S2. Model calculated concentrations of NO₂ (a), O₃ (b), GLY (c), and MGLY (d) in the photolysis cell of the NO₂ convertor. Note that the concentrations at 250 ms represent the condition of the sampled air exits the cell, since the residence time in the cell is about 250 ms.

[Figure]

Figure S3. Model calculated concentrations (green), production rates (red), and destruction rates (blue) of GLY (a) and MGLY (b) in the photolysis cell of the NO$_2$ convertor. Note that the concentrations at 250 ms represent the condition of the sampled air exits the cell, since the residence time in the cell is about 250 ms.

[Figure]

Figure S4. Model calculated concentration (green), production rate (red), and destruction rate (blue) of GLY (a) and MGLY (b) in the photolysis cell of the $NO_2$ converter. The initial concentrations of NMHCs in the model are set to 10 times of the values listed in Table 1. Note that the concentrations at 250 ms represent the condition of the sampled air exits the cell, since the residence time in the cell is about 250 ms.

---

## Author Comment (AC2) · 30 Jun 2019

We thank Referee #2 for his/her comments on our paper, which helped us to improve the quality of the paper. Below, we answer the reviewer's question point by point.

Liu et al. describe a new incoherent broadband cavity enhanced absorption spectrometer (IBBCEAS) for atmospheric measurements of $NO_2$, glyoxal, and methyl glyoxal. The instrument has been thoroughly characterized and performs well compared to existing methods. As a new feature, the instrument used a photolytic converter (NPC) to selectively reduce the amount of $NO_2$ in their sample stream. Major revisions will be required before this paper is acceptable for publication.

(1) There are too many figures in the current draft. Many are not necessary.

**Reply:**

We have revised the manuscript and the number of figures have been shorten to 10. Some figures which we think could be helpful for illustrating the performance of our IBBCEAS system are shown in the Supplement.

(2) Some figures are unclear as the captions are too short.

**Reply:**

Captions of all figures in the manuscript are revised in the way that clearly describing the figure content.

(3) I was not convinced of the merits of the NPC. Is the NPC even necessary? It seems that (valid) data could be obtained without it, though at a slightly higher noise level.

**Reply:**

At the beginning of the experiments, we have the same doubt as you about how much NPC can improve the quality of measurements. Subsequent experiments indicate that NPC not only reduces noise level, but also weakens the measuring interference of high $NO_2$ concentrations. As shown in Fig. 8(a) and (b) in the revised manuscript, the $NO_2$ concentrations reduced by 76% and GLY concentrations reduced by 5% when the NPC was turned on. With respect to well-mixed $NO_2$ and GLY, the concentration of $NO_2$ dropped rapidly while the NPC was running and the concentration of GLY increased. After the NPC turned off, the concentrations of the two compounds returned to the same level as before the NPC was turned on (Fig. 8c). So the concentration of GLY determined by IBBCEAS can be underestimated in the presence of high $NO_2$ concentrations and by utilizing NPC to reduce the $NO_2$ concentration, the measurement accuracy of GLY is improved.

(4) Does the NPC generate glyoxal and methyl glyoxal when ambient air containing organic compounds is sampled? The MCM lists 127 precursors for glyoxal and 143 precursors for methyl glyoxal. I haven't gone through these lists in detail, but it seems that more characterization experiments are warranted in my opinion. It seems to me that the NPC may generate more issues than it solves.

**Reply:**

Given the size of the photolysis cell (i.e., a cylinder with 60 mm length and 13.4 mm

inner diameter) and the sampling flow rate of 2 L/min, the residence time of the sampled air in the cell is about 0.25 s. In such a short period, the production of GLY and MGLY in the cell is negligible which can be verified by model simulations. The model includes the full MCM chemistry (version 3.3.1, http://mcm.leeds.ac.uk/MCM/) for all NMHCs and their oxidation products. The initial concentrations of OH, $HO_2$, $O_3$, NO, HONO, CO, $CH_4$, $C_2$-$C_{12}$ NMHCs are set to the average values obtained during a field observation campaign in 2018 in Yangtze River Delta, China (Table S1). The total OH reactivity due to NMHCs is about 20 $s^{-1}$. The relative humidity, temperature, and pressure are constrained by the campaign averages, i.e., 60%, 300 K, 1007.65 hPa, respectively. The photolysis frequencies are constrained by values calculated from the spectral actinic flux inside the cell (Fig. S1). The spectral actinic flux is estimated by the LED emission spectrum and by the concentration change of NO, $NO_2$, and $O_3$ when $NO_2$ standard (100 ppb) is running through the system. The initial values of $NO_2$, GLY, MGLY are set to 60 ppb, 100 ppt, and 100 ppt, respectively. Fig. S2 shows the calculated concentrations of $NO_2$, $O_3$, GLY, and MGLY for the period of 0.25 s residence time. While the $NO_2$ concentration drops from 60 ppb to 15 ppb leading to an increase of $O_3$ concentration by approximately 45 ppb, the change of GLY and MGLY concentrations is only around 1%. As shown in Fig. S3, the production rate of GLY and MGLY increases to around 5 ppb $h^{-1}$ which could only result in maximumly 0.34 ppt increase of GLY and MGLY. Even if we increase the initial NMHCs concentration in the model by a factor of 10, the maximum production of GLY and MGLY within 250 ms is less than 2 ppt which is only 2% of their initial concentration (Fig. S4). Since the NMHCs concentration in the model represents typical atmospheric condition influenced by urban air pollution, our model simulation clearly shows that the NPC can be used in the field observations for removing $NO_2$ without additional production of GLY and MGLY.

The above discussions are added in the Supplement.

(5) The convolved spectrum of MGLY (Figure 4, thick purple line) does not look correct.

**Reply:**
Thank you for pointing out this mistake. We tested two high resolution cross sections of MGLY at the very beginning of our experiments, one is from Meller et al. (1991) and the other is from Staffelbach et al.(1995). In the manuscript, we mistook convolution results based on Staffelbach's spectrum instead of that based on Meller's spectrum. We revised Fig. 4 accordingly and checked the whole manuscript to ensure that the correct cross sections are used in every part involving MGLY spectral fitting.

Specific comments
pg 2, line 28 "Thalman et al. first encountered the challenge of fitting GLY and MGLY absorption spectra in the presence of high $NO_2$ concentrations (Thalman et al., 2015). To our knowledge, an effective method has not yet to solve this problem." This is a bit of mischaracterization - Thalman et al. [2015] stated that "For both CE-DOAS and BBCEAS ... we do not find significant bias; i.e., an upper-limit change in glyoxal due to

NO$_2$ is derived as +/-200 pptv glyoxal in the presence of 200 ppbv NO$_2$ (or bias of 1 pptv glyoxal/1 ppbv NO$_2$)."- see also major comment #3.
**Reply:**
According to Fig. 4 in Thalman et al. (2015), we can find that the fluctuation of the measured glyoxal concentration increased at least twice when NO$_2$ concentration reached 150 ppb. In addition, even though the upper-limit change in glyoxal due to NO$_2$ is derived as ±200 ppt glyoxal in the presence of 200 ppb NO$_2$ on the whole, the measurement interference on glyoxal already reached around 150 ppt when 50 ppb NO$_2$ was added. Since the ambient mixing ratios of glyoxal typically range between 10 ppt and 100 ppt, and NO$_2$ concentration in polluted area can reach 50 ppb or even higher, accurate quantification of glyoxal could be a challenge under this condition.

pg 7 line 31. Consider adding a digit to each uncertainty as otherwise 4% + 2% = 4%.
**Reply:**
Revised accordingly in the manuscript.

pg 9 line 23. "The presence of high NO$_2$ concentrations may affect the spectral fitting of GLY and MGLY". Perhaps, but this is not obvious from Figure 9 (which seem OK to me). To make their point, it seems like the authors could perform a rather easy experiment: Deliver a constant concentration of GLY (or MGLY) and add increasing amounts of NO$_2$ to this blend. Are the correct GLY (or MGLY) concentrations retrieved?
**Reply:**
Similar experiment as you suggested was performed in the Sect. 4.2. We delivered a constant concentration of GLY and add decreasing amounts of NO$_2$ to this blend for four times. The concentration of GLY measured by IBBCEAS increased as that of NO$_2$ decreased, although only the NO$_2$ concentration was reduced manually. The experimental phenomenon indicates that the presence of NO$_2$ will affect the retrieved concentration of GLY.

Figure 1 (a) Consider adding dimensions to this sketch, such as the inter-mirror distance
**Reply:**
Revised accordingly in the manuscript.

Figure 2 is not necessary in my opinion. Simply state the result in the text.
**Reply:**
Thank you for your advice. Figure 2 is deleted and corresponding results are stated in the text of the revised manuscript.

Figure 3. You may want to state the dimensions of the mixing chamber, temperature of the cold trap, and flow rates.
**Reply:**
Detailed information such as the dimensions of the mixing chamber is added to the caption of Figure 3. This figure is moved to the Supplement.

Figure 4 The convolved methylglyoxal spectrum does not look right.

**Reply:**

Revised accordingly in the manuscript.

Figure 6. This factor will be flow rate dependent, and the sample flow rate should be stated in the caption. Please zoom in on the ratio more (0.85 to 0.95) and state the standard deviation of this measurement. The effective path was determined from a comparison of measured to known concentration; the figure is misleading as it is suggested that it was determined by turning the purge gases on and off (filling the purge flow volumes would take much more time than shown here). Consider clarifying this in the text.

**Reply:**

Thank you for your advice. Figure 6 is re-plotted as you suggested and its caption is rewritten including sample flow rate and standard deviation of this measurement. This figure is moved to Supplement. The method of determining effective path that we used in the manuscript is measuring $NO_2$ concentration with or without purging flows. This kind of method has been approved by peers. For example, Duan et al. (2018) calculated the reduction of the effective cavity length successfully by measuring $O_2$-$O_2$ collision pair with or without purging flows.

Figure 11 The figure caption should describe the experiment conducted here in more detail. Figure 11 (c) The figure caption should indicate what is meant by the yellow overlay. Are the times displayed correct? It seems that the $NO_2$ mixing ratios stay at the high levels well after the NPC was turned on, but increase again before the NPC is turned off?

**Reply:**

The figure caption is rewritten as following and Figure 11 (c) is re-plotted.

"(a) $NO_2$ only test: A concentration gradient of $NO_2$ gas samples were measured twice, with the NPC on (yellow overlay) and off. The removal efficiency is constant at 76 % for different $NO_2$ concentrations. (b) GLY only test: The constant concentration of GLY produced by the standard gas generator was measured with the NPC on (yellow overlay) and off. A small fraction of the GLY (5%) was photolyzed by the NPC. (c) $NO_2$ and GLY mixture test: Well-mixed $NO_2$ and GLY was measured with the NPC on (yellow overlay) and off. The concentration of $NO_2$ dropped while the NPC was running and that of GLY increased. After the NPC was turned off, their concentrations returned to the same level as before the NPC was turned on."

Figure 12 - what does 'fixing' the concentration mean? What is 'first', 'second', 'third', and 'fourth'?

**Reply:**

The figure caption is rewritten as following.

"(a) Keep the concentration of GLY constant and mix it evenly with different concentrations of $NO_2$ for four times. The concentration of GLY measured by IBBCEAS

increased as that of NO$_2$ decreased, although only the NO$_2$ concentration was reduced manually. (b) Simulate a series of spectra with constant GLY concentration and increasing NO$_2$ concentration and then calculate concentrations of GLY and NO$_2$ by retrieving these simulated spectra. The retrieved GLY concentration decreased with the increasing NO$_2$ concentration, although the set value of GLY concentration in the simulation was kept constant."

Figure 13 (b). There are two sets of concentrations displayed which one is methylglyoxal, and what is the other one?
**Reply:**
Both sets of concentrations displayed are methylglyoxal. The figure caption is rewritten as following. This figure is moved to Supplement.
"(a) Four spectral fit ranges (440-451 nm, 445-453 nm, 440-453 nm, and 430-453 nm) for MGLY. (b) Each spectral fit range was used to fit two experimental sets of MGLY produced by the standard gas generator. Experimental results of both sets indicated that fit range has a great influence on the fitted concentrations of MGLY."

Figure 14. The figure caption should indicate what is meant by the yellow overlay. (b) Are the times displayed correct? It seems that the NO$_2$ mixing ratios stay at the high levels well after the NPC was turned on, but increase again before the NPC is turned off? The photo-oxidation of certain organic molecules (e.g., benzene, toluene etc.) can yield glyoxal or methylglyoxal. It looks like this is happening here. Consider evaluating the NPC while sampling an atmospheric background of those VOCs.
**Reply:**
The figure caption is rewritten as following and Figure 14 (b) is re-plotted in the manuscript.
"(a) The constant concentration of MGLY produced by the standard gas generator was measured with the NPC on (yellow overlay) and off. The effect of NPC on MGLY concentration is negligible. (b) Well-mixed NO$_2$ and MGLY was measured with the NPC on (yellow overlay) and off. The concentration of NO$_2$ dropped while the NPC was running and that of MGLY increased. After the NPC was turned off, their concentrations returned to the same level as before the NPC was turned on."
The experimental results shown in Figure 14(b) are the concentration of measuring the mixture of methylglyoxal and NO$_2$. Because the gas mixture only contained methylglyoxal, NO$_2$, and N$_2$, the increase of methylglyoxal concentration cannot be caused by the photooxidation of certain organic molecules. So the methylglyoxal concentration determined by IBBCEAS could be underestimated and we used NPC to reduce the interference.
The advice of evaluating the NPC while sampling an atmospheric background of VOCs is reasonable. The results of model simulation shows that the production of GLY and MGLY in the NPC is negligible. For more information please refer to the reply of major comment (4).

Fig 14 (c) The caption does not adequately describe what is shown in the figure.

**Reply:**

The caption of Fig. 14(c) is rewritten as following.

"(c) Simulate a series of spectra with constant MGLY concentration and increasing $NO_2$ concentration and then calculate concentrations of MGLY and $NO_2$ by retrieving these simulated spectra. The retrieved MGLY concentration decreased with the increasing $NO_2$ concentration, although the set value of MGLY concentration in the simulation was kept constant."

Reference

Duan, J., Qin, M., Ouyang, B., Fang, W., Li, X., Lu, K., Tang, K., Liang, S., Meng, F., Hu, Z., Xie, P., Liu, W., and Häsler, R.: Development of an incoherent broadband cavity-enhanced absorption spectrometer for in situ measurements of HONO and NO2, Atmos Meas Tech, 11, 4531-4543, 2018.

Meller, R., Raber, W., Crowley, J. N., Jenkin, M. E., and Moortgat, G. K.: THE UV-VISIBLE ABSORPTION-SPECTRUM OF METHYLGLYOXAL, Journal of Photochemistry and Photobiology a-Chemistry, 62, 163-171, 1991.

Staffelbach, T. A., Orlando, J. J., Tyndall, G. S., and Calvert, J. G.: THE UV-VISIBLE ABSORPTION-SPECTRUM AND PHOTOLYSIS QUANTUM YIELDS OF METHYLGLYOXAL, J Geophys Res-Atmos, 100, 14189-14198, 1995.

Thalman, R., Baeza-Romero, M., Ball, S., Borrás, E., Daniels, M., Goodall, I., Henry, S., Karl, T., Keutsch, F., and Kim, S.: Instrument intercomparison of glyoxal, methyl glyoxal and NO2 under simulated atmospheric conditions, 2015. 2015.

Table S1. Initial concentration of species included in the model simulation.

| Species | Concentration | Species | Concentration |
|---|---|---|---|
| OH | $10^7$ cm$^{-3}$ | NO | 0.60 ppb |
| HO2 | $10^9$ cm$^{-3}$ | HONO | 0.45 ppb |
| CH4 | 1.9 ppm | CO | 0.33 ppm |
| O3 | 70.00 ppb | SO2 | 1.33 ppb |
| NO2 | 60 ppb | MGLY | 100.0 ppt |
| GLY | 100.0 ppt | TOLUENE | 0.515 ppb |
| C2H2 | 1.000 ppb | NC8H18 | 0.034 ppb |
| CBUT2ENE | 0.050 ppb | EBENZ | 0.140 ppb |
| C2H4 | 0.900 ppb | MXYL | 0.045 ppb |
| C2H6 | 2.130 ppb | NC9H20 | 0.019 ppb |
| IC4H10 | 0.380 ppb | OXYL | 0.076 ppb |
| IC5H12 | 0.330 ppb | STYRENE | 0.017 ppb |
| NC4H10 | 0.650 ppb | IPBENZ | 0.012 ppb |
| NC5H12 | 0.240 ppb | PBENZ | 0.014 ppb |
| PENT1ENE | 0.004 ppb | METHTOL | 0.016 ppb |
| TPENT2ENE | 0.002 ppb | PETHTOL | 0.015 ppb |
| C5H8 | 0.233 ppb | NC10H22 | 0.017 ppb |
| CPENT2ENE | 0.003 ppb | TM135B | 0.014 ppb |
| M22C4 | 0.019 ppb | OETHTOL | 0.036 ppb |
| M23C4 | 0.020 ppb | TM123B | 0.014 ppb |
| M2PE | 0.100 ppb | PXYL | 0.045 ppb |
| M3PE | 0.076 ppb | NC11H24 | 0.019 ppb |
| HEX1ENE | 0.009 ppb | C3H8 | 2.010 ppb |
| NC6H14 | 0.130 ppb | C3H6 | 0.120 ppb |
| M2HEX | 0.030 ppb | C4H6 | 0.005 ppb |
| CHEX | 0.044 ppb | TM124B | 0.016 ppb |
| M3HEX | 0.041 ppb | TBUT2ENE | 0.002 ppb |
| BENZENE | 0.364 ppb | BUT1ENE | 0.040 ppb |
| NC7H16 | 0.055 ppb | | |

[Figure]

Figure S1. Spectral actinic flux inside the photolysis cell of the NO$_2$ convertor.

[Figure]

Figure S2. Model calculated concentrations of $NO_2$ (a), $O_3$ (b), GLY (c), and MGLY (d) in the photolysis cell of the $NO_2$ convertor. Note that the concentrations at 250 ms represent the condition of the sampled air exits the cell, since the residence time in the cell is about 250 ms.

[Figure]

Figure S3. Model calculated concentrations (green), production rates (red), and destruction rates (blue) of GLY (a) and MGLY (b) in the photolysis cell of the NO₂ convertor. Note that the concentrations at 250 ms represent the condition of the sampled air exits the cell, since the residence time in the cell is about 250 ms.

[Figure]

Figure S4. Model calculated concentration (green), production rate (red), and destruction rate (blue) of GLY (a) and MGLY (b) in the photolysis cell of the $NO_2$ converter. The initial concentrations of NMHCs in the model are set to 10 times of the values listed in Table 1. Note that the concentrations at 250 ms represent the condition of the sampled air exits the cell, since the residence time in the cell is about 250 ms.

---

## Author Response (AR2)

We thank Referee #1 for his/her comments on our paper, which helped us to improve the quality of the paper. Below, we answer the reviewer's question point by point.

The authors have addressed my concerns. Couple of suggestions:
Abstract (pg 1, line 27): replace "largely" with a more quantitative statement
**Reply:**
It is difficult to provide a general quantitative statement from this study. We have modified the text as "… the quality of the spectra fitting and the measurement accuracy of ambient GLY and MGLY under $NO_2$ rich environments could be improved."

pg 5, line 28: "Diagram ..." correct grammar
**Reply:**
We revise the text as "The schematic setup of the standard gas generator is shown in Fig. S5 in the Supplement."

We thank Referee #3 for his/her comments on our paper, which helped us to improve the quality of the paper. Below, we answer the reviewer's question point by point.

The manuscript was improved and can be accepted for publication. Some minor suggestions:
Page 10, line 27, a brief description of the model simulation results is suggested to be included in the text.
**Reply:**
We add the following text in the end of Section 2.3.
"Moreover, the photolysis of $NO_2$ and the resulted $O_3$ production in the NPC could probably lead to additional GLY and MGLY production in the condition of high VOC environment. However, as illustrated in the Supplement, this artifact is negligible given the short residence time of the sampled air in the NPC."

Page 11, line 31, the unit of the fitting residual was lost.
**Reply:**
We add the unit of the fitting residual.

[revised manuscript text omitted]